# A Brain Graph Foundation Model: Pre-Training and Prompt-Tuning for Any Atlas and Disorder

## Abstract

As large language models (LLMs) continue to revolutionize AI research, there
is a growing interest in building large-scale brain foundation models to advance
neuroscience. While most existing brain foundation models are pre-trained on
time-series signals or region-of-interest (ROI) features, we propose a novel graph-
based pre-training paradigm for constructing a brain graph foundation model. In
this paper, we introduce the **Brain Graph Foundation Model**, termed **BrainGFM**,
a unified framework that leverages graph contrastive learning and graph masked
autoencoders for large-scale fMRI-based pre-training. BrainGFM is pre-trained
on a diverse mixture of brain atlases with varying parcellations, significantly ex-
panding the pre-training corpus and enhancing the model's ability to generalize
across heterogeneous fMRI-derived brain representations. To support efficient
and versatile downstream transfer, we integrate both graph prompts and language
prompts into the model design, enabling BrainGFM to flexibly adapt to a wide
range of atlases, neurological and psychiatric disorders, and task settings. Further-
more, we employ meta-learning to optimize the graph prompts, facilitating strong
generalization to previously unseen disorders under both few-shot and zero-shot
learning conditions via language-guided prompting. BrainGFM is pre-trained on 27
neuroimaging datasets spanning 25 common neurological and psychiatric disorders,
encompassing 2 types of brain atlases (functional and anatomical) across 8 widely-
used parcellations, and covering over 25,000 subjects, 60,000 fMRI scans, and a
total of 400,000 graph samples aggregated across all atlases and parcellations.

## 1 Introduction

With the rise of large language models (LLMs) Achiam et al. [2023], large-scale pre-trained founda-
tion models (FMs) have been proposed across various domains, including computer vision Touvron
et al. [2023], natural language processing Achiam et al. [2023], and data mining Xia et al. [2024].
Recently, the field of neuroscience has also begun to witness the emergence of brain foundation
models. As a widely used data modality in neuroscience, functional magnetic resonance imaging
(fMRI) Markiewicz et al. [2021], Bycroft et al. [2018] plays a crucial role in understanding brain
function and dysfunction. Developing a fMRI-based brain foundation model is of great importance for
advancing neuroscience and its translational research. Due to the high complexity and cost of fMRI
data acquisition Van Essen et al. [2012], Cui et al. [2022], coupled with strong heterogeneity and
substantial inter-subject variability, most existing traditional deep learning-based fMRI models Wei
et al. [2025], Kan et al. [2022], Jiao et al. [2025] are trained on relatively small datasets. Consequently,
these models are typically tailored to specific tasks, disorders, or cohorts, resulting in limited general-
izability, poor flexibility, and weak transferability to unseen tasks, datasets or disorder conditions.
Furthermore, training on small-scale datasets often leads to under-fitting, ultimately compromising
model performance and reliability. These issues have become common limitations of traditional deep

learning models for fMRI. However, they can be effectively addressed by building fMRI brain FMs. Since FMs are typically pre-trained on large-scale datasets with rich diversity Touvron et al. [2023], Xia et al. [2024], spanning various data types and knowledge representations within the neuroscience domain, the resulting models exhibit strong generalization and broad applicability. As a result, fMRI FMs can be more readily adapted to a wide range of downstream data, tasks and neurological and psychiatric disorders.

Previous fMRI FMs have uniformly adopted Transformer-based architectures and were exclusively pre-trained on either time-series data or ROI-level features, resulting in two main categories: time-series-based Caro et al. [2023], Thomas et al. [2022] and ROI-based Yang et al. [2024], Hu et al. [2024], Dong et al. [2024] fMRI brain FMs. However, building brain FMs faces several critical challenges that many previous approaches have either overlooked or failed to effectively address. **(1). Data Availability & Heterogeneity.** fMRI data are difficult and costly to collect and pre-process Poldrack and Gorgolewski [2017], yet pre-training FMs typically requires large-scale datasets. Existing fMRI datasets are not only limited in quantity but also exhibit substantial heterogeneity across sources. Effectively leveraging and integrating these heterogeneous datasets is thus a fundamental challenge. Many existing brain FMs have constructed relatively large-scale fMRI pre-training datasets, but these are typically based on a single brain parcellation or atlas Thomas et al. [2022]. This overlooks the fact that integrating multiple parcellation templates can not only expand the scale of available fMRI data but also provide diverse and even complementary features across different brain parcellations Hermosillo et al. [2024]. **(2). Pre-Training Computational Cost.** The computational cost of pre-training brain FMs typically depends on the form of the fMRI data and the chosen pre-training strategy. Time-series-based brain FMs are pre-trained directly on raw fMRI time series, resulting in high computational demands with masked modeling pre-training paradigm. While ROI-based brain FMs are more efficient, they often neglect inter-regional connectivity, leading to suboptimal performance on various downstream tasks. Striking a balance between computational efficiency and modeling effectiveness remains a pressing issue. **(3). Adaptability and Generalization for Few/Zero-Shot Transfer.** Pre-trained brain FMs needs to be fine-tuned to various downstream tasks, datasets, atlases and disorders. However, full-parameter fine-tuning is often inefficient, requires large amounts of labeled data, and typically previous brain FMs Caro et al. [2023], Yang et al. [2024] support only one disorder or atlas during the downstream inference. In addition, in many real-world scenarios, downstream tasks may involve new atlases, datasets and disorders unseen during pre-training, with very limited (few-shot) or even no labeled data available (zero-shot). Adapting FMs to such few-shot or zero-shot settings poses a significant yet highly valuable challenge. Most existing brain FMs Caro et al. [2023], Thomas et al. [2022], Dong et al. [2024] have not considered few-shot or zero-shot scenarios, which limits their generalizability and flexibility. These three challenges correspond to four essential aspects of pre-training brain FMs: data collection, model pre-training & fine-tuning, and downstream task adaptation.

**Contributions.** To address the key challenges outlined above and overcome the limitations of prior work, we propose the Brain Graph Foundation Model, named BrainGFM, specifically designed for heterogeneous fMRI data, with a particular focus on graph-based modeling. We propose corresponding solutions within our model to enhance BrainGFM, enabling it to become a more powerful brain FM compared to previous approaches. **(1).** To enable effective pre-training of brain FMs, we construct a large-scale fMRI dataset comprising 27 widely used fMRI datasets. This collection includes over 25,000 subjects, 60,000 fMRI scans, and 25 common neurological and psychiatric disorders. Unlike previous brain foundation models, each fMRI sample in our dataset is processed using 2 different brain functional and anatomical atlases, including 8 parcellations with various resolutions and partitions, significantly increasing the scale and diversity of the data. This also allows the pre-trained model to capture complementary feature representations across multiple parcellations. **(2).** Prior brain FMs have predominantly relied on fMRI time series or ROI-level features for both pre-training and fine-tuning. In this work, we creatively introduce a graph-based backbone for building brain graph FMs. This approach offers the advantage of maintaining computational efficiency comparable to ROI-based FMs, while achieving performance on par with time-series-based FMs. **(3).** To enhance the generalizability and adaptability of the model, we discard conventional fine-tuning and introduce a graph prompt-tuning. Under the multi-task and multi-dataset training paradigm of meta-learning, this approach improves the model's ability to perform few-shot adaptation across diverse tasks and datasets. In addition, we incorporate language prompt tokens, including atlas/parcellation tokens and task/disorder tokens, to guide the pre-trained BrainGFM in adapting to entirely unseen downstream datasets, atlases, tasks, and disorders in zero-shot settings.

## 2 Related Works

### 2.1 Pre-Training Approaches for Brain Foundation Models Using fMRI

The emergence of large-scale foundation models, such as LLMs Achiam et al. [2023], has demonstrated strong potential across various domains. In neuroscience, recent efforts have introduced brain FMs (e.g., using fMRI), which can be broadly classified into time-series-based Dong et al. [2024], Caro et al. [2023], Thomas et al. [2022] and ROI-based models Yang et al. [2024], Hu et al. [2024], both primarily relying on generative pre-training using masked modeling. In contrast to these approaches, our work introduces the first graph-based fMRI foundation model, which leverages the brain's topological structure through graph representations. We incorporate both graph generative pre-training Hou et al. [2022] and graph contrastive pre-training Qiu et al. [2020], Wei et al. [2024], unifying two major paradigms in graph representation learning.

### 2.2 Graph Pre-Training and Prompt Learning

Pre-training is a fundamental step in the development of foundation models, with most approaches categorized into contrastive-based and generative-based paradigms. While graph model pre-training differs from that in vision and language domains, it generally follows these two strategies. To facilitate zero-shot generalization, language prompts Achiam et al. [2023] have been widely used in NLP and vision, providing semantic guidance that enables pre-trained models to adapt to unseen tasks without parameter updates. In contrast, graph prompts have been proposed to address few-shot adaptation in graph neural networks. Inspired by prefix-tuning Li and Liang [2021], graph prompts Sun et al. [2023] introduce a small set of task-specific parameters that can be optimized efficiently while keeping the backbone frozen. This approach improves sample efficiency and reduces computational cost in adapting to new graph-based tasks with limited data.

### 2.3 Meta-Learning

Meta-learning Finn et al. [2017], Hospedales et al. [2021], also known as "learn to learn" aims to train models that can quickly adapt to new tasks using only a small number of labeled examples. It typically involves learning a good initialization or adaptation strategy by optimizing over a distribution of related tasks. Meta-learning has been widely adopted in few-shot learning scenarios and has shown strong potential for improving generalization across tasks and domains Sun et al. [2023]. In our study, meta-learning is employed to train the graph prompt under the few-shot setting, enabling the unification and generalization across diverse brain atlases and neurological disorders.

## 3 Methodology

As illustrated in Figure 1, we propose BrainGFM, a graph-based paradigm that distinguishes itself from previous time series-based and ROI-based brain FMs. Our framework consists of four main stages: large-scale fMRI graph data collection and pre-processing, graph pre-training for building our brain graph foundation model, multi-task meta-learning optimization for few-shot learning, and graph/language prompt-tuning for zero-shot adaption.

### 3.1 Construction of Large-Scale fMRI Pre-Training Dataset

***Motivation.*** *Brain parcellations with different resolutions and partitions offer complementary representations of brain structure and function, and different disorders may be best characterized under different parcellations.*

As shown in Figure 1(a), we curated a large-scale fMRI dataset by aggregating 27 widely used fMRI datasets from different sites and institutions, covering 25 common neurological and psychiatric disorders. Unlike existing brain FMs, our dataset incorporates fMRI data processed using 8 parcellations, including Schaefer100/Schaefer200/Schaefer300 Schaefer et al. [2018], AAL116/AAL3v1 Tzourio-Mazoyer et al. [2002], SHEN268 Shen et al. [2013], Power264 Power et al. [2011], and Gordon333 Gordon et al. [2016]. For each subject, we extracted raw fMRI time series using these brain atlases and constructed fMRI brain graphs by computing and binarizing the Pearson correlation between time series among brain ROIs. Integrating multiple atlases allows us to expand the dataset to eight

times the size of using a single parcellation, enabling more diverse representations and facilitating the learning of atlas-invariant brain patterns. The inclusion of multiple atlases not only increases the diversity and volume of the training data but also enables the model to learn parcellation-specific features, significantly enhancing the generalization and robustness of the pre-trained BrainGFM. Note that detailed information regarding the benchmark settings, including task types, dataset splits, neurological disorder categories, and atlas/parcellation choices, can be found in *Appendix* N and P.

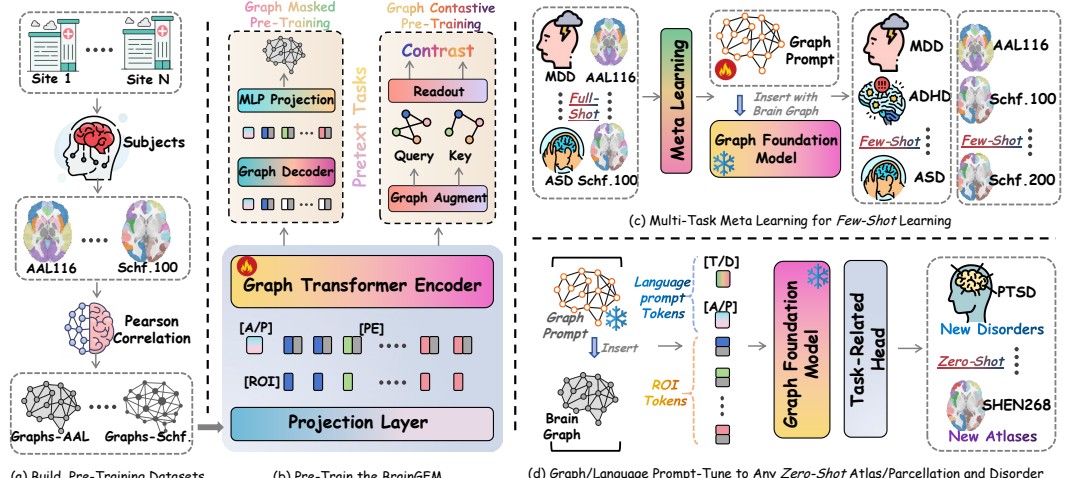

Figure 1: The pipeline of our proposed BrainGFM. (a) A large-scale brain fMRI graph dataset is constructed for pre-training. (b) BrainGFM is pre-trained using graph contrastive and masked autoencoder strategies, with atlas/parcellation tokens [A/P] to encode atlas-specific information. (c) Instead of full fine-tuning, we introduce graph prompts and use meta-learning to optimize them for few-shot adaptation, keeping the graph FM backbone frozen. (d) Finally, we freeze both the model and graph prompts, and use language prompts to enable zero-shot transfer to unseen tasks. Note that "Schf." means Schaefer atlas.

## 3.2 Graph Pre-Training for Building BrainGFM

***Motivation.*** *The graph foundation model approaches the* *effectiveness* *of time-series-based foundation models, while matching the* *efficiency* *of ROI-based foundation models.*

We adopt a Graph Transformer Yun et al. [2019] as the backbone of our BrainGFM. As illustrated in Figure 1(b), we first transform the input fMRI brain graphs and project them to obtain brain graph embeddings, where each token corresponds to a brain ROI. We apply `Positional Encoding Tokens`, denoted as `[PE]`, to each brain ROI, enabling the model to perceive and learn the topological and spatial characteristics of each ROI. Unlike conventional positional encodings used in standard Transformer models Vaswani et al. [2017], graph-based positional encodings are inherently different, as they require encoding the relative positions between nodes in the graph structure. Compared to the commonly used Laplacian positional encoding Dwivedi et al. [2023] and node degree positional encoding You et al. [2019] in graph-based models, we adopt a more efficient alternative Random Walk Structural Encoding (RWSE) Dwivedi et al. [2021] as our positional encoding strategy. More details about PEs can be found in *Appendix* F. Furthermore, inspired by language models in NLP, we insert `Atlas/Parcellation Tokens`, denoted as `[A/P]`, to the brain graph embeddings during pre-training, enabling the model to better distinguish and learn from different atlases and parcellations. Note that the construction of [A/P] tokens is described in detail in Section 3.4. Incorporating this token enables the model to capture parcellation-specific patterns, which is crucial as prior studies Hermosillo et al. [2024], Liu et al. [2023], Wu et al. [2025] show that different brain disorders are better represented by specific parcellations. For instance, MDD benefits from Schaefer200 or Power264 and ASD is better captured by Shen268 or Schaefer200. Embedding such parcellation-aware information helps improve model generalization across disorders and atlas configurations.

We follow the graph pre-training paradigm to pre-train our BrainGFM. To fully leverage the potential of graph pre-training, we adopt two widely used pretext tasks in graph domain: *graph contrastive*

174 *learning (GCL)* You et al. [2020] pre-training and *graph masked autoencoders (GMAE)* Hou et al.
175 [2022] pre-training. For GCL pre-training, we apply graph augmentation to the fMRI brain graphs
176 by randomly dropping nodes and edges to generate positive and negative pairs of queries and keys.
177 The contrastive loss is then computed by contrasting the positive and negative graph pairs. For
178 GMAE pre-training, we randomly mask nodes and edges in the input brain graphs to obtain masked
179 brain graphs. These graphs are then passed through a graph autoencoder with an encoder-decoder
180 architecture to reconstruct the masked nodes and edges, optimized using a mean squared error (MSE)
181 loss. Note that both GCL and GMAE pre-training share the same encoder, which serves as the core
182 of our BrainGFM, which enables the encoder of BrainGFM to benefit from both contrastive and
183 generative paradigms, resulting in a more robust and well-pre-trained backbone. More details about
184 these two graph pre-training methods can be found in *Appendix* J and K.

### 3.3 Few-Shot Graph Prompt-Tuning via Meta Learning Optimization

186 ***Motivation.*** *The graph prompt, optimized via multi-task meta-learning, enables the fully frozen graph*
187 *foundation model to be effectively adapted to new, unseen tasks under few-shot settings.*

188 As illustrated in Figure 1(c), after completing the pre-training stage, we need to fine-tune the pre-
189 trained FM to various downstream tasks, including different atlases and disorders. However, for
190 fMRI data, traditional full-parameter fine-tuning faces two major limitations. **(1).** The collection of
191 fMRI data for neurological and psychiatric disorders is often time-consuming and labor-intensive,
192 and for some rare diseases, only a very limited number of samples are available. When performing
193 full-parameter fine-tuning on a large-scale foundation model with limited data, the optimization of
194 model parameters becomes insufficient, leading to significant performance degradation. **(2).** Full-
195 parameter fine-tuning requires substantial training time and computational resources, making it less
196 practical in resource-constrained environments. Therefore, we introduce graph prompts Sun et al.
197 [2023] to prompt-tune to our BrainGFM to different diseases and atlases. Following prior work on
198 graph prompt learning, we design brain graph prompts specifically for brain graphs, with a structure
199 consistent with the input brain graphs. Each node in the graph prompt is a learnable parameter, and
200 the collection of all nodes forms a learnable vector set. For the edges, we initialize a fully learnable
201 edge matrix, where each entry is also trainable. This design allows the graph prompt to flexibly adapt
202 the FM to various downstream tasks without modifying the backbone parameters.

203 To optimize the parameters of our brain graph prompts, we introduce meta-learning to train the
204 graph prompts. Specifically, we construct a multi-task dataset in which each task corresponds to
205 a different brain disorder and atlas pair. By adopting this meta-learning paradigm, the optimized
206 graph prompts can be flexibly transferred to unseen diseases and atlases, enabling effective adaptation
207 using only a small number of samples from the few-shot downstream tasks. During the meta-learning
208 optimization process, all parameters of the pre-trained model are kept frozen, and only the graph
209 prompt parameters, which are relatively lightweight, are updated. This design enables fast tuning
210 and adaptation. Moreover, the few-shot sample setting is particularly well-suited for optimizing the
211 small number of graph prompt parameters; in contrast, using limited samples to fine-tune a large
212 pre-trained foundation model with numerous parameters would lead to insufficient training and severe
213 underfitting. The task/disorder-specific features related to each disorder, atlas, or parcellation are
214 thus captured and stored entirely within the well-trained brain graph prompts. As a result, with the
215 help of the learned task-specific graph prompts, the frozen BrainGFM remains consistently ready to
216 be efficiently and rapidly adapted via prompt-tuning to unseen tasks, datasets, disorders and atlases,
217 even when only a few samples (few-shot settings) are available. More details about the meta-learning
218 datasets split and training procedure can be found in the Appendix I 8.

### 3.4 Zero-Shot Graph/Language Prompt-Tuning

220 ***Motivation.*** *The language prompt guides the frozen pre-trained graph foundation model and meta-*
221 *learned graph prompt to achieve effective zero-shot transfer across diverse disorders and atlases.*

222 Building on the few-shot capability, we further introduce language prompts to enable more generalized
223 zero-shot learning by jointly guiding both the graph prompt and the pre-trained foundation model.
224 In zero-shot scenarios, the parameters of the graph prompt are also frozen, meaning the model
225 cannot rely on prompt adaptation through learning. Instead, the language prompt provides semantic
226 guidance, allowing the model to generalize and adapt to unseen downstream data, tasks, and disorder

types without any gradient-based updates. As shown in Figure 1(d), in order to enable the model to recognize and distinguish between different tasks and disorders in zero-shot settings, we introduce Task/Disorder Tokens, denoted as [T/D], during the downstream fine-tuning stage, following a similar design with Atlas/Parcellation Tokens [A/P]. To construct the [T/D] tokens, we first generate a textual description for each disorder, including its full name, abbreviation, and a concise clinical summary. For example, for Major Depressive Disorder, the corresponding text description is: *"Major Depressive Disorder (MDD) is a common mental illness characterized by persistent and profound low mood, loss of interest, and cognitive impairment, significantly affecting daily life and social functioning."* Otte et al. [2016] We then encode these textual descriptions using a BERT model Devlin et al. [2019] pre-trained on large-scale medical corpora, such as BioClinicalBERT Huang et al. [2019], Alsentzer et al. [2019], to obtain semantic-rich text embeddings. These embeddings are subsequently projected and embedded as [T/D] tokens, which are incorporated into the model during downstream adaptation. Similarly, the construction of [A/P] tokens is also based on language text. For each atlas and parcellation, we provide a textual description of its name, such as "Schaefer100", "Schaefer200", or "AAL116". These text descriptions are then encoded using the BioClinicalBERT pre-trained model to extract language embeddings, which are subsequently transformed into [A/P] tokens. As shown in Figure 1(d), the [T/D] and [A/P] tokens are ultimately concatenated with the ROI tokens from the graph embeddings as **language prompt tokens**. This combined input is then fed into the foundation model to guide feature extraction specific to the given dataset, task, and disorder. By introducing disorder-specific semantic priors through the [T/D] and [A/P] tokens, the model is better equipped to capture characteristics from different tasks, disorders, atlas and parcellations, thereby improving its downstream adaption ability in zero-shot settings without any training.

## 4 Experiments

### 4.1 Comparison with Other Methods

***Datasets.*** To demonstrate the superiority of our BrainGFM, we conducted comparative experiments. Specifically, 10 common types of neurological and psychiatric disorders were selected from a total of 25 disorders, spanning 6 datasets among the 27 datasets we collected. More details about benchmarks and datasets can be found in Appendix N. ***Baselines.*** We compare our method against a series of baseline models. Based on the data representation type, these baselines are categorized into three groups: time-series-based methods, ROI-based methods, and graph-based methods. Based on the training paradigm, they are divided into two groups: non-pre-trained FMs and pre-trained FMs. All pre-trained models are retrained on our collected pre-training dataset to ensure a fair comparison. More details about baselines can be found in Appendix M. ***Metrics.*** We evaluate all methods using four metrics: AUC, accuracy (ACC), sensitivity (SEN), and specificity (SPE). More detailed information on disorders, datasets, and benchmarks can be found in the supplementary material. As shown in Table 1, our method outperforms all previous approaches and achieves state-of-the-art performance. The pre-trained FMs significantly outperforms models without pre-training. Our method, built upon a graph transformer backbone, substantially surpasses ROI-based brain FMs (BrainMass and BrainNPT), and also outperforms time-series brain FMs (BrainLM).

### 4.2 Ablation Study for Full/Few/Zero-Shot on Graph/Language Prompt and Meta Learning

Figure 2 illustrates the classification accuracy under four different data regimes, Full-Shot (100%), Few-Shot (10%), Few-Shot (1%), and Zero-Shot (0%), across three representative downstream datasets: ABIDE II, ADHD 200, and ADNI 2. We observe a consistent performance degradation across all methods as the available training data decreases, with the largest performance gap occurring under the most data-scarce setting (Zero-Shot). Vanilla Models, which lack any form of pre-training, perform the worst across all settings, highlighting their limited generalization ability. Introducing the FM (BrainGFM) without graph prompts leads to notable performance improvements, confirming the effectiveness of graph-based pre-training. The inclusion of graph prompts (FM + G-Prompt) further enhances accuracy, particularly in Few-Shot and Zero-Shot regimes, indicating their role in injecting structural prior knowledge. When combined with meta-learning (FM + G-Prompt + Meta L.), the model demonstrates increased adaptability and robustness under limited supervision. Finally, incorporating language prompts (FM + G-Prompt + Meta L. + Lan. Prompt) consistently achieves the best performance across all datasets and data regimes, underscoring the benefit of semantic guidance

Table 1: Comparison among different methods on 10 brain disorders on Schaefer100 atlas. Pink indicates the best performance.

| Method | Pre-Trained | ADHD200 (ADHD) | | | | ABIDE II (ASD) | | | | ADNI 2 (AD) | | | | HBN (MDD) | | | | HBN (ANX) | | | |
|---|---|---|---|---|---|---|---|---|---|---|---|---|---|---|---|---|---|---|---|---|---|
| | | AUC | ACC | SEN | SPE | AUC | ACC | SEN | SPE | AUC | ACC | SEN | SPE | AUC | ACC | SEN | SPE | AUC | ACC | SEN | SPE |
| Vanilla GCN | ✗ | 61.6 | 63.7 | 59.0 | 63.3 | 63.6 | 65.6 | 60.7 | 63.3 | 68.4 | 70.2 | 64.3 | 72.2 | 70.2 | 71.8 | 68.9 | 71.7 | 75.6 | 78.2 | 79.6 | 72.4 |
| BrainGNN | ✗ | 60.5 | 63.3 | 63.1 | 57.3 | 61.2 | 62.7 | 57.3 | 65.2 | 69.1 | 70.6 | 66.9 | 73.4 | 68.8 | 73.0 | 71.4 | 65.7 | 76.3 | 77.2 | 72.7 | 80.5 |
| Vanilla TF | ✗ | 62.4 | 64.6 | 60.8 | 62.1 | 65.3 | 65.2 | 64.2 | 61.9 | 71.7 | 74.4 | 68.6 | 74.2 | 74.1 | 76.6 | 71.3 | 78.7 | 77.8 | 80.9 | 81.6 | 73.9 |
| Graph TF | ✗ | 64.6 | 65.3 | 62.1 | 66.3 | 65.2 | 67.1 | 67.7 | 63.6 | 73.3 | 76.6 | 70.4 | 75.9 | 76.4 | 77.8 | 73.1 | 80.2 | 80.5 | 81.8 | 77.9 | 83.5 |
| BrainNetTF | ✗ | 63.3 | 64.6 | 65.2 | 61.1 | 66.5 | 66.6 | 66.9 | 65.7 | 74.3 | 76.4 | 77.2 | 70.7 | 75.7 | 74.9 | 72.6 | 78.5 | 78.4 | 81.1 | 75.5 | 80.9 |
| BrainNPT | ✓ | 62.3 | 66.5 | 61.6 | 59.2 | 65.5 | 67.3 | 65.8 | 71.6 | 70.6 | 75.7 | 65.2 | 75.3 | 72.8 | 74.1 | 68.5 | 76.7 | 75.6 | 77.2 | 79.1 | 71.4 |
| BrainLM | ✓ | 66.3 | 68.6 | 62.7 | 70.4 | 68.5 | 70.2 | 65.6 | 71.1 | 76.7 | 81.5 | 72.4 | 80.5 | 75.1 | 81.9 | 81.9 | 69.4 | 82.3 | 83.6 | 77.6 | 87.2 |
| BrainMass | ✓ | 65.5 | 66.1 | 63.6 | 69.9 | 67.3 | 68.5 | 69.3 | 64.8 | 76.6 | 79.3 | 71.5 | 80.1 | 75.7 | 78.5 | 82.2 | 74.4 | 79.8 | 82.7 | 79.0 | 80.6 |
| BrainGFM (Ours) | ✓ | 70.6 | 72.2 | 67.3 | 73.4 | 71.2 | 73.5 | 70.4 | 69.8 | 80.3 | 85.1 | 76.2 | 84.4 | 83.6 | 85.5 | 85.8 | 77.9 | 85.2 | 86.3 | 87.7 | 82.6 |

| Method | Pre-Trained | HBN (OCD) | | | | HBN (PTSD) | | | | SubMex_CUD (CUD) | | | | UCLA_CNP (SCHZ) | | | | UCLA_CNP (BP) | | | |
|---|---|---|---|---|---|---|---|---|---|---|---|---|---|---|---|---|---|---|---|---|---|
| | | AUC | ACC | SEN | SPE | AUC | ACC | SEN | SPE | AUC | ACC | SEN | SPE | AUC | ACC | SEN | SPE | AUC | ACC | SEN | SPE |
| Vanilla GCN | ✗ | 69.7 | 76.9 | 61.3 | 76.6 | 77.5 | 78.7 | 82.1 | 71.8 | 63.8 | 66.2 | 55.6 | 72.5 | 77.2 | 79.5 | 72.3 | 81.8 | 64.3 | 69.6 | 70.1 | 58.2 |
| BrainGNN | ✗ | 70.1 | 74.6 | 76.7 | 66.4 | 75.2 | 78.0 | 69.3 | 81.9 | 62.5 | 64.8 | 67.1 | 57.5 | 78.1 | 77.5 | 83.7 | 72.6 | 63.8 | 67.3 | 58.5 | 69.2 |
| Vanilla TF | ✗ | 72.5 | 78.8 | 64.5 | 78.2 | 78.2 | 80.1 | 71.7 | 85.9 | 65.6 | 66.7 | 60.4 | 71.1 | 76.8 | 80.2 | 71.9 | 80.4 | 65.1 | 70.6 | 69.4 | 61.7 |
| Graph TF | ✗ | 73.6 | 81.5 | 66.7 | 77.5 | 77.1 | 81.6 | 74.3 | 80.2 | 67.5 | 69.8 | 59.4 | 75.7 | 78.9 | 80.8 | 75.1 | 81.6 | 67.4 | 70.2 | 70.5 | 64.3 |
| BrainNetTF | ✗ | 74.7 | 80.4 | 79.2 | 69.1 | 79.4 | 81.9 | 74.6 | 84.6 | 66.8 | 67.5 | 61.7 | 71.4 | 75.1 | 77.3 | 73.5 | 77.9 | 68.5 | 71.6 | 73.4 | 64.8 |
| BrainNPT | ✓ | 71.2 | 75.3 | 63.6 | 78.4 | 76.4 | 80.6 | 70.3 | 82.5 | 63.7 | 64.2 | 57.5 | 69.3 | 75.0 | 76.5 | 68.7 | 81.9 | 65.5 | 68.7 | 60.2 | 70.1 |
| BrainLM | ✓ | 78.2 | 83.3 | 72.6 | 84.5 | 80.2 | 84.8 | 81.5 | 73.6 | 68.9 | 71.3 | 63.2 | 73.1 | 81.6 | 82.1 | 77.4 | 85.5 | 70.3 | 73.5 | 74.7 | 66.3 |
| BrainMass | ✓ | 76.8 | 81.2 | 80.6 | 72.5 | 78.1 | 82.5 | 73.2 | 82.4 | 67.5 | 69.4 | 71.7 | 64.0 | 80.8 | 81.2 | 76.5 | 84.2 | 69.7 | 72.3 | 65.8 | 73.3 |
| BrainGFM (Ours) | ✓ | 80.4 | 85.8 | 86.7 | 78.5 | 83.2 | 86.3 | 79.5 | 87.4 | 71.1 | 74.6 | 67.7 | 75.5 | 84.2 | 86.7 | 80.4 | 87.9 | 73.5 | 76.3 | 69.6 | 78.2 |

in enabling zero-shot generalization. These results collectively validate the synergistic contribution of these techniques in building a flexible and generalizable brain FMs.

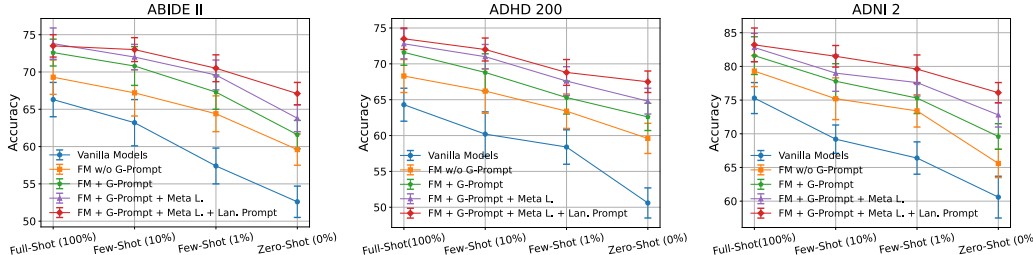

Figure 2: Performance comparison across different settings (Full-Shot, Few-Shot, Zero-Shot) on three datasets: ABIDE II, ADHD 200, and ADNI 2. The results demonstrate the progressive performance gains achieved by incorporating graph prompts (G-Prompt), meta-learning (Meta L.), and language prompts (Lan. Prompt) into the FM (BrainGFM), especially in few-shot and zero-shot settings.

## 4.3 Experiments on Different Atlases and Parcellations with Different ROI Resolutions

To systematically assess the effectiveness of different pre-trained models across a variety of brain atlases and parcellations, we conducted comprehensive ablation studies. Specifically, we fine-tuned four models on fMRI datasets spanning 2 representative atlases and 8 parcellation schemes. The four models include: a vanilla graph transformer trained from scratch; BrainGFM (Functional), pre-trained on the functional Schaefer100 atlas; BrainGFM (Anatomical), pre-trained on the anatomical AAL116 atlas; and BrainGFM (Mixed), pre-trained on a combination of Schaefer100 and AAL116 data. The atlases used in our experiments comprise both functional (Schaefer, SHEN, Power, and Gordon) and anatomical (AAL) types. Among these, the Schaefer atlas provides three resolutions (100, 200, and 300 parcels), and the AAL atlas includes both AAL116 and AAL3v1 parcellations. As illustrated in Figure 3(a), BrainGFM (Functional) outperforms BrainGFM (Anatomical) when evaluated on functional atlases, while the reverse is true for anatomical atlases. In all cases, both types of pre-trained BrainGFM models significantly outperform the vanilla graph transformer trained from scratch, highlighting the benefits of graph pre-training. Notably, although atlas-specific pre-training offers substantial improvements, the BrainGFM (Mixed) model, pre-trained jointly on both functional and anatomical data, achieves the best performance across all downstream atlases. We hypothesize that this superior generalization stems from the complementary nature of anatomical structures and functional connectivity patterns, which jointly enable the model to capture a richer and more diverse set of neurobiological representations.

Overall, as summarized in Figure 3(b), the relative performance of the four models follows two consistent patterns depending on the type of downstream atlas. For functional atlases, BrainGFM (Mixed) performs best, followed by BrainGFM (Functional), BrainGFM (Anatomical), and finally the vanilla model. In contrast, when evaluated on anatomical atlases, the best performance is again achieved by BrainGFM (Mixed), followed by BrainGFM (Anatomical), BrainGFM (Functional), and lastly the vanilla model. These findings underscore the value of incorporating both anatomical and functional information during pre-training to enhance the generalizability of brain graph models across diverse parcellation schemes.

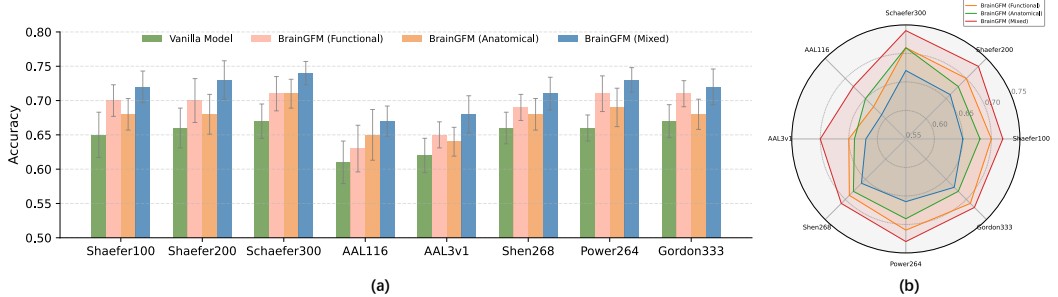

Figure 3: The performance of models pre-trained on different atlases varies across downstream atlases. The experiments are conducted on ABIDE II dataset for ASD classification.

## 4.4 Comparison Among Time-Series, ROI and Graph-Based Foundation Models

We compare four types of brain FMs: time-series-based FM (e.g., BrainLM), ROI-based FM (e.g., BrainMass), vanilla graph-based FM, and our proposed graph-based model, BrainGFM. The comparison spans five key dimensions: model performance, pre-training and fine-tuning efficiency, memory usage, and model complexity.

In terms of performance, BrainLM achieves the best results on AUC and ACC due to its direct modeling of raw fMRI time series, effectively capturing both temporal and spatial patterns. BrainMass, which relies on static ROI features without modeling inter-regional interactions, performs the worst. The vanilla graph-based model shows intermediate performance by explicitly modeling ROI connectivity. BrainGFM, which incorporates fMRI-specific enhancements such as graph prompts and structural encodings, significantly outperforms the vanilla graph model and matches or exceeds the performance of time-series-based models. For computational efficiency, ROI-based models are the fastest in both pre-training and fine-tuning, given their compact input and lack of spatiotemporal modeling. Time-series models are the slowest due to the cost of processing long, high-dimensional sequences. Graph-based models, including

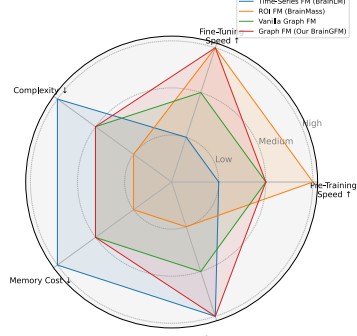

Figure 4: Comparison of performance and efficiency across different brain FMs.

BrainGFM, lie in between. Notably, BrainGFM achieves fast fine-tuning via prompt tuning while maintaining pre-training efficiency similar to the vanilla graph FM, surpassing even ROI-based models in fine-tuning speed. Regarding resource consumption, time-series models are the most memory- and compute-intensive. ROI-based models are the most lightweight. Graph-based models, while slightly more demanding than ROI-based ones due to edge computations, remain significantly more efficient than time-series models. Overall, this evaluation highlights the trade-offs between different brain modeling paradigms and how input representations, time series, ROI features, or graphs, affect both the effectiveness and efficiency of large-scale brain FMs.

## 4.5 Ablation Study on Pre-Training with Different Atlases and Parcellations

To investigate and demonstrate the impact of different atlases and parcellations on the performance of the pre-trained model, we conducted ablation experiments using pre-training datasets constructed from various types of atlases and parcellations. Specifically, we categorized the pre-training datasets into five representative groups: **(1)** a dataset based on a single func-

tional atlas and a single parcellation (Schaefer100), **(2)** a dataset based on a single anatomical atlas and a single parcellation (AAL116), **(3)** a mixed dataset combining both functional and anatomical atlases (Schaefer100 + AAL116), **(4)** a dataset based on a single atlas but incorporating multiple resolutions of parcellations (Schaefer100+200+300), and **(5)** a fully mixed dataset comprising various atlases and parcellations (All 5 Atlases with 8 Parcellations). As shown in Table 2, pre-training on datasets with a single-resolution parcellation reveals that functional atlases, such as Schaefer, outperform anatomical atlases, such as AAL116. This highlights that functional-based atlases are more effective in capturing disease-specific features in the diagnosis of neurological and psychiatric disorders. Additionally, pre-training on datasets mixing different parcellation resolutions within a single atlas (e.g., Schaefer 100+200+300) achieves comparable per-

Table 2: Effect of different atlases on pre-training (ABIDE II, ASD).

| Corpus | Atlas | Parcel. | FT Acc. |
|---|---|---|---|
| w/o Pre-train | - | - | 65.2 / 67.1 |
| Schaefer100 | Func. | Single | 67.5 / 70.2 |
| AAL116 | Anat. | Single | 66.6 / 69.2 |
| Sch(100+200+300) | Func. | Mixed | 68.5 / 71.3 |
| Sch100 + AAL116 | Mixed | Single | 68.8 / 71.6 |
| All Atlases | Mixed | Mixed | 70.5 / 73.3 |

formance to pre-training on datasets combining multiple atlases with one parcellation each (e.g., Schaefer100 + AAL116). Finally, pre-training on datasets that incorporate multiple atlases and parcellations achieves substantially better performance than all previous settings. This improvement can be attributed to the model's ability to comprehensively learn features captured by different atlases, thereby acquiring knowledge from diverse medical and biological perspectives. In addition, the model benefits from learning across parcellations with varying resolutions, which enables it to capture the brain's feature distributions at both global and local scales.

### 4.6 Ablation Study on Different Foundation Model Pre-Training Methods

As shown in Figure 5, we compare the effectiveness of different graph pre-training strategies, including graph contrastive learning (GCL), graph-masked autoencoders (GMAE) and their sequential combination. The results demonstrate that GCL slightly outperforms GMAE, and that combining GCL and GMAE yields further performance gains compared to using either method alone.

Graph contrastive learning (GCL) pre-training primarily focuses on capturing global representations of brain graphs by encouraging the model to aggregate holistic graph-level features and distinguish between different graph attributes and categories. In comparison, graph masked autoencoders (GMAE) pre-training emphasizes

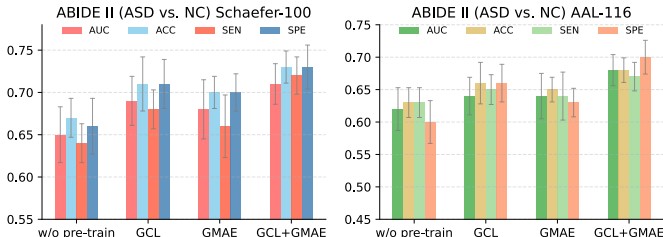

Figure 5: Performance of different graph pre-training methods.

the learning of local representations, where the model reconstructs masked brain ROIs based on information from their local neighborhoods, thereby promoting specialization in ROI-level feature extraction. By sequentially combining GCL and GMAE during pre-training, BrainGFM is able to simultaneously acquire both global and local discriminative capabilities. Notably, the integration of global and local information has been widely recognized as critical for understanding brain organization and pathology in neuroscience and neuroimaging studies. Consequently, our pre-trained model benefits from this multi-scale representation learning, leading to enhanced transferability and improved performance across various downstream tasks.

## 5 Conclusion

We propose BrainGFM, a graph-based brain foundation model pre-trained on heterogeneous fMRI brain graphs constructed from diverse atlases and parcellation schemes. To enhance its generalization and adaptability, we introduce a meta-learning framework to optimize graph prompts, enabling robust few-shot learning under limited data. In addition, we incorporate language prompt tokens to guide zero-shot generalization, allowing BrainGFM to transfer effectively across unseen datasets, tasks, atlases, and neurological disorders. Our large-scale, multi-atlas fMRI dataset provides a rich and diverse training corpus, and BrainGFM demonstrates superior performance in both effectiveness and efficiency compared to prior time-series-based and ROI-based foundation models.

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

# A Simplified Training Pipeline of BrainGFM

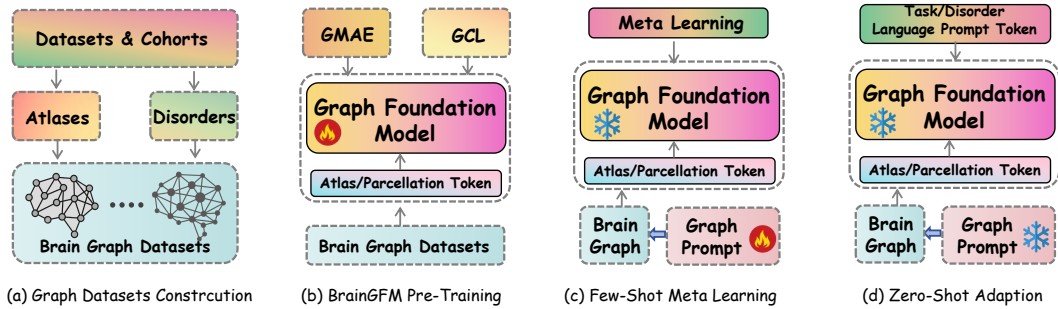

(a) Graph Datasets Constrcution   (b) BrainGFM Pre-Training   (c) Few-Shot Meta Learning   (d) Zero-Shot Adaption

Figure 6: The simplified training pipeline of BrainGFM, covering (a) fMRI graph construction for pre-training, (b) BrainGFM pre-training, (c) meta-learning for few-shot scenarios, and (d) zero-shot adaptation via language prompts.

# B Contributions of BrainGFM for Unifying Cohorts, Atlases, and Disorders

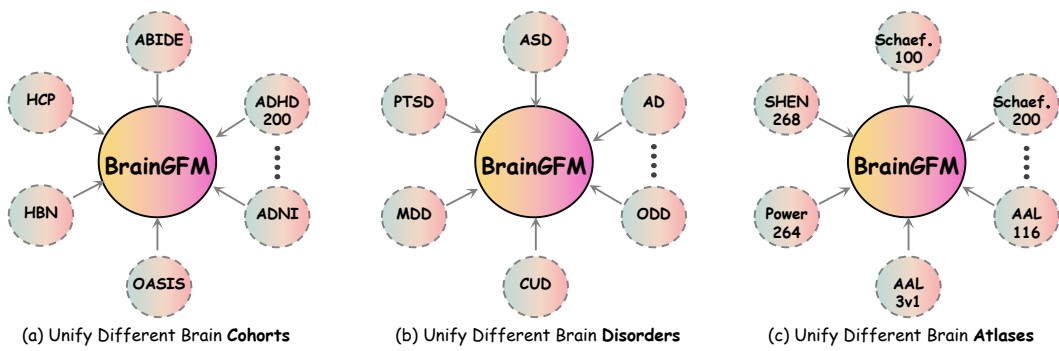

(a) Unify Different Brain **Cohorts**   (b) Unify Different Brain **Disorders**   (c) Unify Different Brain **Atlases**

Figure 7: Our BrainGFM achieves unification in the fMRI domain across three key dimensions: (a) diverse brain datasets and cohorts, (b) multiple neurological and psychiatric disorders, and (c) various brain atlases and parcellations.

# C Comparison of Our Vanilla Graph FM and BrainGFM with Prior Time-Series-based and ROI-based Brain FMs

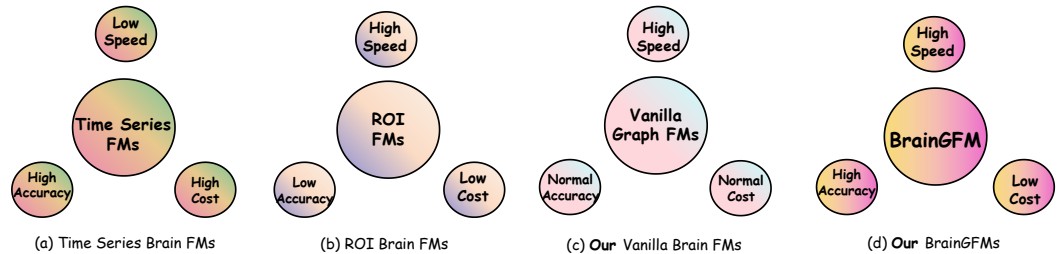

(a) Time Series Brain FMs   (b) ROI Brain FMs   (c) **Our** Vanilla Brain FMs   (d) **Our** BrainGFMs

Figure 8: We compare different brain foundation models in terms of performance, inference speed, and computational cost. The results show that Graph FM provides a trade-off between performance and efficiency compared to Time-Series FM, while our BrainGFM achieves the best overall performance across all aspects.

# D   Ablation Study on Different Tuning Methods

As shown in Table 3, after completing model pre-training, we explored three downstream adaptation strategies: full fine-tuning Sun et al. [2022], parameter-efficient fine-tuning (PEFT) Ding et al. [2023], and graph prompt-tuning Sun et al. [2023]. Full fine-tuning updates all model parameters during downstream training, offering strong performance but at a high computational cost. In contrast, PEFT methods reduce training overhead by modifying only a small subset of parameters or introducing lightweight modules. We specifically evaluate two popular PEFT variants: prefix-tuning and LoRA. Note that the details of all tuning methods are summarized in Table 4.

Graph prompt-tuning further improves efficiency by freezing the entire pre-trained model and updating only a small set of learnable prompt vectors injected into the input. This strategy allows the model to adapt without altering its core parameters, making it highly suitable for resource-constrained settings. In full-shot scenarios, fine-tuning delivers the best results, but PEFT methods achieve competitive performance with significantly lower computational demands. Graph prompt-tuning, while slightly less accurate, offers the best efficiency–adaptability trade-off by minimizing trainable parameters. Given the structural nature of brain graphs, where node and edge features capture complex spatial and relational dependencies, we examine two prompt insertion mechanisms: addition ("+") and multiplication ("×"). Results show that multiplicative insertion consistently outperforms the additive version, likely because scaling features better preserves relational patterns. Moreover, to further leverage topological information, we extend the prompt-tuning framework by incorporating edge prompts in addition to node prompts. This design grants the model greater flexibility to adjust local connectivity, leading to improved transfer performance across diverse downstream tasks.

Table 3: Comparison of Different Tuning Methods on ABIDE II (ASD Classification).

| Tuning Method | FT Flops | Full-Shot FT |
|---|---|---|
| w/o Pre-Training | Very High | 65.2 / 67.1 |
| Fine-Tuning | High | 70.5 / 73.3 |
| PEFT (Prefix) | Low | 69.3 / 72.1 |
| PEFT (LoRA) | Low | 70.6 / 72.6 |
| G Prompt-Tuning (+) | Very Low | 67.4 / 68.7 |
| G Prompt-Tuning (x) | Very Low | 70.1 / 72.6 |
| G Prompt-Tuning w/ Edge (x) | Very Low | 71.2 / 73.5 |

Table 4: Overview of Tuning Methods Evaluated. Fine-tuning updates all model weights; PEFT strategies reduce trainable parameters by introducing lightweight modules; graph prompt-tuning updates only learnable prompts while freezing the backbone.

| Tuning Method | Trainable Parameters | Backbone Frozen | Extra Module | Description |
|---|---|---|---|---|
| Full Fine-Tuning | All | No | No | Updates all weights during downstream training. |
| PEFT (Prefix-Tuning) | Small prefix vectors | Yes | Prefix vectors | Injects trainable tokens into the input sequence. |
| PEFT (LoRA) | Low-rank matrices | Yes | LoRA adapters | Adds trainable rank-decomposed projections to attention layers. |
| Graph Prompt-Tuning (+) | Small prompt vectors | Yes | Prompt tokens | Adds prompts to node features via element-wise addition. |
| Graph Prompt-Tuning (×) | Small prompt vectors | Yes | Prompt tokens | Injects prompts via feature-wise multiplication. |
| G Prompt-Tuning w/ Edge (×) | Node + edge prompts | Yes | Node + edge prompts | Extends prompt injection to edge features for better adaptation. |

# E   Brain fMRI Graph Construction from fMRI Time Series

To construct brain graphs from resting-state fMRI data, we follow a correlation-based approach that captures functional interactions between brain regions. Specifically, for each subject, we extract the regional mean time series $\{\mathbf{t}_i \in \mathbb{R}^T\}_{i=1}^N$ from $N$ brain regions of interest (ROIs), where $T$ is the number of time points. We then compute the Pearson correlation coefficient between every pair of ROI time series:

$$\mathbf{A}_{ij} = \frac{\mathrm{Cov}(\mathbf{t}_i, \mathbf{t}_j)}{\sigma(\mathbf{t}_i) \cdot \sigma(\mathbf{t}_j)} \in [-1, 1],$$

where $\mathrm{Cov}(\cdot, \cdot)$ denotes the covariance and $\sigma(\cdot)$ the standard deviation. The resulting symmetric matrix $\mathbf{A} \in \mathbb{R}^{N \times N}$ serves as the weighted adjacency matrix of the brain graph, representing functional connectivity strengths.

To construct the node features, we reuse the correlation profile of each ROI as its functional embedding. That is, for node $i$, we define its feature vector as:

$$\mathbf{x}_i = \mathbf{A}_{i,:} \in \mathbb{R}^N,$$

which encodes the functional relationships between ROI $i$ and all other ROIs. The resulting graph $\mathcal{G} = (\mathcal{V}, \mathcal{E}, \mathbf{A}, \mathbf{X})$ is fully connected and characterized by node features $\mathbf{X} = [\mathbf{x}_1^\top; \ldots; \mathbf{x}_N^\top] \in \mathbb{R}^{N \times N}$, and edge weights given by $\mathbf{A}$.

# F  Comparison of Positional Encoding Methods

We investigate three commonly used positional encoding (PE) strategies for graph neural networks: *Laplacian Positional Encoding (LPE)*, *Node Degree Positional Encoding*, and *Random Walk Structural Encoding (RWSE)*. Below, we define each method and evaluate their characteristics in the context of fMRI-based brain graph modeling.

**(1) Laplacian Positional Encoding.**  LPE leverages the eigenstructure of the graph Laplacian to capture global graph geometry. The symmetric normalized Laplacian is defined as:

$$\mathbf{L} = \mathbf{I} - \mathbf{D}^{-1/2} \mathbf{A} \mathbf{D}^{-1/2},$$

where $\mathbf{A} \in \mathbb{R}^{N \times N}$ is the adjacency matrix, $\mathbf{D}$ is the degree matrix with $D_{ii} = \sum_j A_{ij}$, and $\mathbf{I}$ is the identity matrix. The PE is obtained by taking the first $k$ non-trivial eigenvectors of $\mathbf{L}$:

$$\mathbf{PE}_{\text{LPE}} = [\mathbf{u}_1, \mathbf{u}_2, \ldots, \mathbf{u}_k],$$

where each $\mathbf{u}_i \in \mathbb{R}^N$ is the $i$-th eigenvector.

*Analysis.* LPE captures rich global topology, but computing eigenvectors is expensive and unstable for large or dynamic graphs. It also suffers from a lack of cross-graph consistency, which limits its effectiveness for pre-training and transfer tasks.

**(2) Node Degree Positional Encoding.**  This method encodes each node $v_i$ with its degree:

$$\mathbf{PE}_{\text{Degree}}(v_i) = \deg(v_i) = \sum_{j=1}^{N} A_{ij},$$

and optionally normalized:

$$\mathbf{PE}_{\text{Degree}}(v_i) = \frac{\deg(v_i)}{\max_j \deg(v_j)}.$$

*Analysis.* Degree encoding is fast and simple, requiring no matrix operations. However, it only captures local connectivity and lacks the expressiveness needed for distinguishing complex structural roles in brain graphs.

**(3) Random Walk Structural Encoding (RWSE).**  RWSE encodes multi-scale structural roles by computing the probability that a random walk returns to the same node at different steps. Define the one-step transition probability matrix:

$$\mathbf{P} = \mathbf{D}^{-1} \mathbf{A},$$

then the $t$-step return probability for node $v_i$ is:

$$\mathbf{PE}_{\text{RWSE}}(v_i) = \left[ \left(\mathbf{P}^1\right)_{ii}, \left(\mathbf{P}^2\right)_{ii}, \ldots, \left(\mathbf{P}^T\right)_{ii} \right].$$

*Analysis.* RWSE is computationally efficient, avoids spectral decomposition, and encodes multi-hop recurrence statistics that are particularly well-suited for the hierarchical and modular structure of brain graphs. Empirically, it provides the most favorable balance between accuracy and scalability.

As shown in Table 5, we evaluate different positional encoding strategies on the ABIDE II dataset for ASD classification. The model without any positional encoding is the fastest but performs poorly. Applying graph pre-training substantially improves performance across all variants, validating its effectiveness. Laplacian PE yields competitive accuracy but suffers from high computational cost due to eigen-decomposition. Node Degree Encoding is computationally efficient but underperforms compared to RWSE. Notably, RWSE achieves the highest performance while maintaining fast inference speed. These results indicate that RWSE offers the best trade-off between accuracy and efficiency, making it the most effective and scalable positional encoding method for our BrainGFM framework.

Table 5: Comparison of Positional Encoding Strategies on ABIDE II (ASD Classification)

| [PE] Type | Pre-Trained | Efficiency | Performance (ACC / AUC) |
|---|---|---|---|
| w/o [PE] | ✗ | Very Fast | 65.2 / 67.1 |
| w/o [PE] | ✓ | Very Fast | 69.5 / 71.7 |
| Laplacian [PE] | ✓ | Slow | 69.2 / 71.3 |
| Node Degree [PE] | ✓ | Fast | 68.4 / 70.5 |
| RWSE [PE] | ✗ | Fast | 66.1 / 68.0 |
| RWSE [PE] | ✓ | Fast | **70.5 / 73.3** |

## G  Unified Training on Brain Graphs with Varying Numbers of Nodes

To handle brain graphs with varying numbers of nodes, we introduce a prompt-based unification mechanism that aligns all graphs to a fixed size $N_{\max}$. Given a graph with $N_i$ nodes ($N_i \leq N_{\max}$), its node feature matrix $\mathbf{X}_i \in \mathbb{R}^{N_i \times F}$ is first augmented with random walk structural encoding (RWSE), and then zero-padded to obtain $\mathrm{Pad}(\mathbf{X}_i) \in \mathbb{R}^{N_{\max} \times F}$.

To inject inductive biases, we employ a learnable node prompt matrix $\mathbf{P} \in \mathbb{R}^{N_{\max} \times F}$, and perform element-wise fusion as:
$$\tilde{\mathbf{X}}_i = \mathrm{Pad}(\mathbf{X}_i) \odot \mathbf{P},$$
where $\odot$ denotes the Hadamard product. To further guide the model, we prepend the task/disorder token [T/D] $\mathbf{x}_{\mathrm{TD}}$ and the atlas/parcellation token [A/P] $\mathbf{x}_{\mathrm{AP}}$, resulting in the full input matrix:
$$\mathbf{Z}_i = [\mathbf{x}_{\mathrm{TD}}; \mathbf{x}_{\mathrm{AP}}; \tilde{\mathbf{X}}_i] \in \mathbb{R}^{(N_{\max}+2) \times F}.$$

Simultaneously, the original adjacency matrix $\mathbf{A}_i \in \mathbb{R}^{N_i \times N_i}$ is expanded to $\hat{\mathbf{A}}_i \in \mathbb{R}^{(N_{\max}+2) \times (N_{\max}+2)}$ by fully connecting the two prompt tokens to all other nodes:
$$\hat{\mathbf{A}}_i = \begin{bmatrix} \mathbf{1}_{2 \times 2} & \mathbf{1}_{2 \times N_{\max}} \\ \mathbf{1}_{N_{\max} \times 2} & \mathrm{Pad}(\mathbf{A}_i) \end{bmatrix}.$$

An attention mask $\mathbf{M}_i \in \{0, 1\}^{N_{\max}+2}$ is also constructed, where $\mathbf{M}_i[j] = 1$ indicates that position $j$ corresponds to a padded node. This mask is used to prevent the self-attention mechanism from attending to invalid positions, ensuring consistency across variable-sized graphs.

To apply the attention mask, we modify the raw attention score matrix computed by the self-attention mechanism:
$$\mathbf{S}_i = \frac{\mathbf{Q}_i \mathbf{K}_i^\top}{\sqrt{d}},$$
where $\mathbf{Q}_i = \mathbf{Z}_i \mathbf{W}_Q$ and $\mathbf{K}_i = \mathbf{Z}_i \mathbf{W}_K$ are the query and key projections of the input matrix $\mathbf{Z}_i$.

The attention mask $\mathbf{M}_i$ is broadcast across the attention heads and used to mask out the scores corresponding to padded nodes by replacing them with $-\infty$:
$$\tilde{\mathbf{S}}_i[j, k] = \begin{cases} \mathbf{S}_i[j, k], & \text{if } \mathbf{M}_i[k] = 0, \\ -\infty, & \text{if } \mathbf{M}_i[k] = 1. \end{cases}$$

This masked score matrix $\tilde{\mathbf{S}}_i$ is then passed through the softmax function to obtain the final attention weights:
$$\mathrm{Attention}(\mathbf{Q}_i, \mathbf{K}_i, \mathbf{V}_i) = \mathrm{Softmax}(\tilde{\mathbf{S}}_i) \cdot \mathbf{V}_i,$$
where $\mathbf{V}_i = \mathbf{Z}_i \mathbf{W}_V$ is the value projection. This ensures that attention is only distributed among valid (unpadded) nodes and the two prompt tokens, making the model robust to input graphs with varying node counts.

## H  Graph Prompt Construction and Insertion.

To unify diverse brain graphs with variable node counts and enable flexible adaptation, we construct a learnable *graph prompt* denoted as $\mathbf{P} \in \mathbb{R}^{N_{\max} \times F}$, where $N_{\max}$ is the maximum number of nodes

across all graphs and $d$ is the feature dimension after RWSE augmentation. The prompt serves as a set of element-wise multiplicative masks for node-wise feature modulation.

Given a brain graph with input node features $\mathbf{X} \in \mathbb{R}^{N \times F}$, where $F' = F_{\text{raw}} + F_{\text{rwse}}$. Then, the graph prompt is applied via element-wise multiplication:

$$\hat{\mathbf{X}} = \mathbf{X} \odot \mathbf{P}$$

where $\odot$ denotes Hadamard (element-wise) product. This modulated feature tensor $\hat{\mathbf{X}}$ is projected into the model hidden space via a learnable projection layer:

$$\mathbf{H}_0 = \text{Proj}(\hat{\mathbf{X}}) \in \mathbb{R}^{N_{\max} \times F_{\text{model}}}$$

## I  Meta-Learning for Unifying Diverse Atlases and Disorders

To support cross-disorder and cross-atlas generalization, we design a meta-learning framework that optimizes only the graph prompt module while keeping the entire pre-trained backbone $\mathcal{F}_\theta$ frozen. The goal is to learn a prompt initialization that can quickly adapt to any new brain graph classification task defined by varying disease types and brain parcellations.

Each task $\mathcal{T}_i$ corresponds to a unique combination of a brain disorder and an atlas (e.g., MDD + Schaefer100, ADHD + AAL116). Given a task $\mathcal{T}_i$, we split its data into a support set $\mathcal{D}_i^{\text{train}}$ and a query set $\mathcal{D}_i^{\text{test}}$.

**Inner Loop: Prompt Adaptation on Single Task.**  We perform task-specific adaptation using the support set by updating only the prompt parameters $\mathcal{P}_\phi$, while keeping the encoder $\mathcal{F}_\theta$ fixed:

$$\phi_i' = \phi - \alpha \nabla_\phi \mathcal{L}_{\mathcal{T}_i}^{\text{train}} \left( \mathcal{F}_\theta(\mathcal{P}_\phi, \mathcal{D}_i^{\text{train}}) \right) \tag{1}$$

This update reflects how the prompt adapts to a particular disorder-atlas context, without altering the pre-trained backbone.

**Outer Loop: Meta-Update Across Tasks.**  We update the prompt initialization $\mathcal{P}_\phi$ by minimizing the query set losses across a batch $B$ of tasks:

$$\phi \leftarrow \phi - \beta \sum_{i=1}^{B} \nabla_\phi \mathcal{L}_{\mathcal{T}_i}^{\text{test}} \left( \mathcal{F}_\theta(\mathcal{P}_{\phi_i'}, \mathcal{D}_i^{\text{test}}) \right) \tag{2}$$

This outer-loop update encourages the learned prompt to generalize across diverse tasks, each characterized by a different disorder and parcellation, while the encoder remains frozen throughout.

---

**Algorithm 1:** Meta-Learning for Graph Prompt Tuning (Frozen Backbone)

---

**Input:** Frozen backbone $\mathcal{F}_\theta$, task set $\{\mathcal{T}_i\}_{i=1}^N$, learning rates $\alpha, \beta$
**Output:** Meta-learned graph prompt parameters $\phi$
Initialize prompt parameters $\mathcal{P}_\phi$ ;
**while** *not converged* **do**
    Sample a batch of tasks $\{\mathcal{T}_i\}_{i=1}^B$ ;
    /* Each task $\mathcal{T}_i$ = (disorder, atlas) pair                  */
    /* Inner Loop:  Adapt prompt on single task               */
    **for** *each task $\mathcal{T}_i$* **do**
        Split into support $\mathcal{D}_i^{\text{train}}$ and query $\mathcal{D}_i^{\text{test}}$ ;
        Compute task-specific adapted prompt:
        $\phi_i' = \phi - \alpha \nabla_\phi \mathcal{L}_{\mathcal{T}_i}^{\text{train}}(\mathcal{F}_\theta(\mathcal{P}_\phi, \mathcal{D}_i^{\text{train}}))$ ;
        /* Note:  $\mathcal{F}_\theta$ is frozen, only $\phi$ of $\mathcal{P}_\phi$ is updated     */
    /* Outer Loop:  Update shared prompt using query losses       */
    $\phi \leftarrow \phi - \beta \sum_i \nabla_\phi \mathcal{L}_{\mathcal{T}_i}^{\text{test}}(\mathcal{F}_\theta(\mathcal{P}_{\phi_i'}, \mathcal{D}_i^{\text{test}}))$ ;
**return** $\phi$

---

# J  Details of Graph Masked Autoencoder (GMAE) and Graph Contrastive Learning (GCL)

We propose a unified pre-training framework that combines Graph Masked Autoencoder (GMAE) and Graph Contrastive Learning (GCL) to learn robust and generalizable representations for brain graphs. These two components complement each other: GMAE enables fine-grained feature-level recovery through generative reconstruction, while GCL encourages invariance under perturbations by contrasting different views of the same graph.

## J.1  Graph Masked Autoencoder (GMAE)

Inspired by recent advances in masked autoencoding Hou et al. [2022], we apply random masking to both nodes and edges of the input graph. Given a graph with $N$ nodes, adjacency matrix $\mathbf{A} \in \mathbb{R}^{N \times N}$, and node features $\mathbf{X} \in \mathbb{R}^{N \times F}$, we randomly sample a subset of nodes $\mathcal{V}_M \subset \mathcal{V}$ to mask. For each masked node $v_i \in \mathcal{V}_M$, its input feature is replaced with a learnable mask token $\mathbf{x}_{[M]} \in \mathbb{R}^D$. The masked node feature matrix $\tilde{\mathbf{X}}$ is defined as:

$$\tilde{\mathbf{x}}_i = \begin{cases} \mathbf{x}_{[M]}, & \text{if } v_i \in \mathcal{V}_M \\ \mathbf{x}_i, & \text{otherwise} \end{cases}$$

To further increase the learning difficulty, we also apply structural masking by dropping edges in the adjacency matrix. Specifically, each edge is dropped independently with probability $p$, resulting in a corrupted adjacency matrix:

$$\tilde{\mathbf{A}} = \mathbf{A} \odot \mathbf{M}_e$$

where $\mathbf{M}_e \in \{0,1\}^{N \times N}$ is a symmetric binary mask sampled from a Bernoulli distribution with parameter $1 - p$, and $\odot$ denotes element-wise multiplication.

The GMAE encoder processes the corrupted graph $(\tilde{\mathbf{X}}, \tilde{\mathbf{A}})$ and produces latent embeddings, which are then passed to a lightweight decoder to reconstruct the original node features of the masked nodes. The reconstruction objective is defined as:

$$\mathcal{L}_{\text{rec}} = \frac{1}{|\mathcal{V}_M|} \sum_{v_i \in \mathcal{V}_M} \|\hat{\mathbf{x}}_i - \mathbf{x}_i\|_2^2$$

where $\hat{\mathbf{x}}_i$ is the predicted feature from the decoder.

## J.2  Graph Contrastive Learning (GCL)

In parallel with the generative pathway, we apply contrastive learning to enforce view-invariant representations. Specifically, we generate two augmented views of the same input graph by applying lightweight stochastic perturbations (i.e., random feature dropout and edge dropout). One view is treated as the `query`, while the other serves as the `key`. Let $(\mathbf{X}^{(q)}, \mathbf{A}^{(q)})$ and $(\mathbf{X}^{(k)}, \mathbf{A}^{(k)})$ denote the two views; these are passed through a shared encoder to obtain corresponding representations $z_q$ and $z_k$. We adopt the NT-Xent contrastive loss Qiu et al. [2020] to maximize the similarity between matching query-key pairs from the same graph while distinguishing them from others in the batch:

$$\mathcal{L}_{\text{CL}} = -\frac{1}{B} \sum_{b=1}^{B} \log \frac{\exp(\text{sim}(z_q^{(b)}, z_k^{(b)})/\tau)}{\sum_{b'=1}^{B} \mathbb{1}_{[b' \neq b]} \exp(\text{sim}(z_q^{(b)}, z_k^{(b')})/\tau)}$$

where $\text{sim}(\cdot, \cdot)$ denotes cosine similarity, $\tau$ is a temperature hyperparameter, and $B$ is the batch size. Here, $b$ and $b'$ index different samples within the batch, where each sample corresponds to an augmented graph.

By encouraging the embeddings of different augmented views of the same graph to be aligned, the model learns representations that are invariant to small perturbations in node features and topology, thereby improving generalization across downstream tasks.

# K    Comparison between Two Graph Pre-Training Methods

Table 6 provides a comprehensive comparison between Graph Contrastive Learning (GCL) and Graph Masked Autoencoder (GMAE) as two prominent pre-training paradigms for graph neural networks. GCL emphasizes learning discriminative representations by contrasting positive and negative graph pairs, making it particularly effective for classification and retrieval tasks. In contrast, GMAE focuses on reconstructing masked parts of the graph, encouraging the model to capture fine-grained structural details and local contextual information. While GCL tends to produce compact and abstract embeddings that distinguish samples globally, GMAE yields richer and more structure-aware representations suitable for reconstruction and local reasoning. However, both approaches have limitations: GCL's effectiveness heavily depends on the design of graph augmentations, potentially neglecting subtle local cues; GMAE, on the other hand, is sensitive to the masking ratio and may underperform in tasks requiring global discrimination. These complementary characteristics suggest that combining both strategies may lead to more robust and generalizable graph representations.

Table 6: Comparison between Graph Contrastive Learning (GCL) and Graph Masked Autoencoder (GMAE) Pre-training.

|  | **Graph Contrastive Learning (GCL)** | **Graph Masked Autoencoder (GMAE)** |
|---|---|---|
| **Main Objective** | Learn to pull together positive pairs and push apart negative pairs, focusing on discriminative representations. | Learn to reconstruct masked node/edge features, focusing on structure-aware and fine-grained representations. |
| **Feature Focus** | Emphasizes global discriminative features that distinguish between different samples. | Emphasizes local structure awareness and detailed pattern recovery. |
| **Pre-training Strategy** | Contrastive loss (e.g., InfoNCE) between augmented graph views. | Masking parts of the graph and reconstructing them via a decoder. |
| **Best for** | Classification, retrieval, tasks requiring strong discrimination. | Reconstruction, generation, local reasoning, and also benefiting classification. |
| **Learning Tendency** | Learns compact, abstract representations that excel at distinguishing samples. | Learns rich, detailed representations that capture local and global graph structures. |
| **Potential Drawbacks** | Sensitive to augmentation design; may overlook fine-grained local details. | Sensitive to masking ratio; may focus too much on local patterns without sufficient global discrimination. |

# L    Implement Details

During pre-training, we set the batch size to 128 and used the Adam optimizer with a learning rate of 0.0001. The number of training epochs was set to 100. For downstream classification tasks, we set the batch size to 16 in the full-shot setting, and to 1 in both the few-shot and zero-shot settings. The Adam optimizer was also used for these tasks, with the same learning rate of 0.0002. In the meta-learning setup, we trained the model for 30 epochs. More settings about pre-training and graph transformer backbone and meta learning can be found in Table 7, 8, 10, 9.

# M    Baselines

The Table 11 provides a systematic comparison of various brain foundation models and baseline methods across multiple dimensions, including architectural type, data domain, pre-training strategy, and tuning method. These approaches can be broadly categorized into three groups: conventional models without pre-training (e.g., Vanilla GCN Kipf and Welling [2016], BrainGNN Li et al. [2021], Vanilla ROI-based Tansformer (TF) Yun et al. [2019]), ROI- or time-series-based pre-trained models (e.g., BrainNPT Hu et al. [2024], BrainMass Yang et al. [2024], BrainLM Caro et al. [2023]), and our proposed graph-based foundation model, BrainGFM.

The conventional models do not leverage any pre-training and are limited to single parcellation and single-disorder settings, resulting in restricted generalization capabilities. ROI-based models such as BrainNPT and BrainMass employ generative pre-training on region-level features, enabling improved

Table 7: Training and architectural hyperparameters used in the `Graph Transformer Backbone`.

| Parameter | Value | Description |
| --- | --- | --- |
| `batch_size` | 128 | Number of samples per batch during training. |
| `learning_rate` | 0.0001 | Learning rate used by the optimizer. |
| `GMAE_decoder_layers` | 4 | Number of layers in the decoder of the Graph Masked Autoencoder (GMAE). |
| `ff_hidden_size` | 256 | Hidden dimension of the feed-forward layer in the Transformer. |
| `num_classes` | 2 | Number of output classes (mainly used for downstream classification tasks). |
| `num_self_att_layers` | 4 | Number of Transformer self-attention layers used in the encoder. |
| `dropout` | 0.3 | Dropout rate used for regularization. |
| `num_GNN_layers` | 4 | Number of GNN layers stacked in the encoder. |
| `nhead` | 8 | Number of attention heads in each multi-head self-attention layer. |
| `hidden_dim` | 128 | Dimensionality of hidden representations in the encoder. |
| `max_feature_dim` | 512 | Maximum input node feature dimension after projection. |
| `rwse_steps` | 5 | Number of steps in random walk positional encoding. |

Table 8: Training hyperparameters used in the `MAML-style Meta-Learning Framework`.

| Parameter | Value | Description |
| --- | --- | --- |
| `meta_epochs` | 50 | Number of meta-training epochs (outer loop iterations). |
| `meta_batch_size` | 8 | Number of tasks sampled per meta-update step. |
| `inner_steps` | 1 | Number of inner-loop gradient update steps on each task. |
| `inner_lr` | 0.0002 | Learning rate used in the inner loop (task-specific adaptation). |
| `outer_lr` | 0.0001 | Learning rate used in the outer loop (meta-model update). |
| `k_folds` | 5 | Number of folds used in task-specific K-Fold data splitting. |
| `support_set_size` | 80% | Number of samples used for inner-loop training (support set), determined by fold split. |
| `query_set_size` | 20% | Number of samples used for outer-loop meta-update (query set), determined by fold split. |
| `task_sampling` | Random | Strategy used to sample tasks from the training pool per meta-iteration. |

performance across a limited number of disorders. BrainLM further enhances temporal modeling through time-series-based generative pre-training.

In contrast, BrainGFM adopts both generative and contrastive graph-based pre-training strategies and supports multiple adaptation paradigms, including full fine-tuning and graph prompt-tuning. It is the only model capable of handling full-shot, few-shot, and zero-shot scenarios. Trained on multiple parcellations and evaluated across 25 disorders, BrainGFM demonstrates superior generalizability and adaptability. Overall, it stands out as the only comprehensive brain FMs that supports structural graph modeling, multi-task transfer, cross-parcellation generalization, and versatile tuning strategies.

# N  Benchmarks, Datasets, Disorders and Downstream Tasks

Table 12 provides a comprehensive summary of the datasets used in our framework, categorized into four functional groups: **Pre-train**, **Internal Test**, **Semi-External Test**, and **External Test**. This partitioning is designed to systematically evaluate the performance and generalization ability of our brain FMs across varying levels of domain similarity. The **Pre-train** group includes 19 datasets

Table 9: Common support/query set splits used in MAML-style meta-learning.

| Support:Query | Typical Use Case | Description |
|---|---|---|
| 50% : 50% | Balanced Learning | Equal emphasis on adaptation and generalization. Often used when data size is sufficient. |
| 66% : 34% | Stronger Adaptation | More data is allocated to support the inner loop updates. Suitable for tasks with high variability. |
| 33% : 67% | Stronger Generalization | Emphasis on generalization performance, especially useful when measuring transferability. |

Table 10: Settings and considerations for the number of inner loop steps in MAML-style meta-learning.

| Inner Steps | Typical Scenario | Description |
|---|---|---|
| 1 | Fast Adaptation | Common choice with low compute cost. Provides basic task adaptation and supports batched meta-updates. |
| 3 | Balanced Trade-off | Provides stronger task-specific learning while maintaining reasonable training cost. Often used in practice. |
| 5 | Enhanced Adaptability | Allows deeper inner adaptation. Useful for complex or highly diverse tasks, but increases overfitting risk. |
| >5 | Rarely Used | Risk of overfitting support set and high computational cost. Not commonly used unless thoroughly validated. |

comprising over 50,000 samples from both healthy individuals and patients diagnosed with a broad spectrum of neurological and psychiatric disorders, such as Alzheimer's disease (AD), mild cognitive impairment (MCI), ADHD, ASD, major depressive disorder (MDD), post-traumatic stress disorder (PTSD), and substance use disorder (CUD). These datasets provide rich and diverse training samples for learning a robust and generalizable representation. The **Internal Test** group is constructed from a subset of the pre-training datasets and is used to evaluate in-distribution performance, where both the disorders and acquisition protocols are seen during pre-training. This setting assesses how well the model fits to familiar domains. The **Semi-External Test** group includes datasets involving diseases that overlap with the pre-training stage but originate from different sites, scanners, or cohort distributions. This setting simulates moderate domain shifts and is used to measure the model's transferability to partially unseen distributions. Finally, the **External Test** group consists of datasets that are entirely excluded from pre-training and validation stages, containing distinct population sources and clinical conditions. This group serves as a stringent benchmark for zero-shot generalization, testing the model's ability to adapt to entirely new domains, disorders, and acquisition protocols. Overall, this structured dataset split enables a rigorous and hierarchical evaluation of the model's robustness, transfer performance, and zero-shot generalization capabilities across a wide range of real-world neuroimaging scenarios.

Table 11: Comparison of brain foundation models and baselines across different architectural types, domains, pre-training strategies, and tuning methods.

| Model | Foundation | Model Type | Domain | Pre-Training Method | Tuning Method | Tuning Shot | Parcellation | Disorder |
|---|---|---|---|---|---|---|---|---|
| Vanilla GCN | ✗ | Graph | Spatial | - | - | - | Single | Single |
| BrainGNN | ✗ | Graph | Spatial | - | - | - | Single | Single |
| Vanilla TF | ✗ | ROI | Spatial | - | - | - | Single | Single |
| Graph TF | ✗ | Graph | Spatial | - | - | - | Single | Single |
| BrainNetTF | ✗ | ROI | Spatial | - | - | - | Single | Single |
| BrainNPT | ✓ | ROI | Spatial | ROI Generative | Fine-Tuning | Full/Few-shot | Single | Multiple (< 5) |
| BrainMass | ✓ | ROI | Spatial | ROI Generative | Fine-Tuning | Full/Few-shot | Single | Multiple (10) |
| BrainLM | ✓ | Time Series | Temporal | Time Series Generative | Fine-Tuning | Full-shot | Single | Multiple (< 5) |
| BrainGFM | ✓ | Graph | Spatial | Graph (Gener. + Contra.) | Fine/Prompt-Tuning | Full/Few/Zero-shot | Multiple | Multiple (25) |

Table 12: Overview of Neuroimaging Datasets Used for Pre-Training and Evaluation. We group datasets by their function in our pipeline: pre-training, internal test, semi-external test, and external test. The table lists dataset names, number of unique subjects, total samples, and associated disorders.

| Function | Datasets | Source | Subjects | Samples | Disease |
|---|---|---|---|---|---|
| **Pre-train** | ABCD | Casey et al. [2018] | 11,878 | 35,770 | Multiple |
| | ADHD 200 | consortium [2012] | 973 | 1382 | ADHD |
| | ABIDE I | Di Martino et al. [2014] | 1,112 | 1,112 | ASD |
| | ADNI 3 | Jack Jr et al. [2008] | 1,071 | 1,410 | AD, MCI |
| | AOMIC | Snoek et al. [2021] | 210 | 210 | Multiple |
| | AURORA | McLean et al. [2020] | 284 | 284 | PTSD |
| | CAM_CAN | Shafto et al. [2014] | 652 | 652 | - |
| | CATD | Nielson et al. [2023] | 127 | 454 | Multiple |
| | GSP | Holmes et al. [2015] | 1,569 | 2,706 | - |
| | HCP-Aging | Bookheimer et al. [2019] | 724 | 724 | - |
| | EMBARC | Trivedi et al. [2016] | 308 | 308 | MDD |
| | LEMON | Babayan et al. [2019] | 213 | 213 | MDD |
| | HABS | Dagley et al. [2017] | 284 | 1,371 | - |
| | PREVEND_AD | Tremblay-Mercier et al. [2021] | 343 | 2,427 | AD |
| | SRPBS_Japan | Yamashita et al. [2019] | 1,410 | 1,410 | ASD |
| | NYU_CUD | Kelly et al. [2011] | 29 | 56 | CUD |
| | OASIS3 | LaMontagne et al. [2019] | 1,172 | 4,090 | Dementia |
| | HBN | Alexander et al. [2017] | 2,228 | 4,039 | Multiple |
| | SubMex_RTMS | Angeles-Valdez et al. [2024] | 150 | 150 | CUD |
| **Internal Test** | ADHD 200 | consortium [2012] | 973 | 1382 | ADHD |
| | HBN | Alexander et al. [2017] | 2,282 | 4,039 | Multiple |
| | OASIS3 | LaMontagne et al. [2019] | 1,172 | 4,090 | Dementia |
| **Semi-External Test** | ABIDE II | Di Martino et al. [2014] | 1,044 | 1,044 | ASD |
| | ADNI 2 | Weiner et al. [2013] | 1,171 | 1,306 | AD, MCI |
| | SubMex_CUD | Angeles-Valdez et al. [2022] | 135 | 135 | CUD |
| **External Test** | UCLA_CNP | Poldrack et al. [2016] | 261 | 261 | Multiple |
| | REST-META-MDD | Yan et al. [2019] | 2,379 | 2,379 | MDD |

Table 13 and 14 summarizes the 25 brain disorder classification tasks selected from various public neuroimaging datasets. These tasks span a remarkably broad range in terms of disease types, age groups, and data sources, reflecting the diversity and complexity of real-world clinical scenarios. The downstream evaluation covers a wide spectrum of brain conditions, including neurodevelopmental disorders (e.g., ADHD, ASD, SLD), affective and emotional disorders (e.g., MDD, anxiety, bipolar disorder), psychotic disorders (e.g., schizophrenia), neurodegenerative diseases (e.g., AD, MCI, dementia), and substance use disorders (e.g., CUD). Subject age ranges from as young as 5 years old (e.g., ABIDE II, HBN) to elderly adults nearing 90 years old (e.g., ADNI, OASIS3), capturing the full human lifespan from brain development to cognitive decline. To ensure scientific rigor and fairness, we carefully constructed sex-balanced subsets for all labeled downstream tasks, meaning that each task includes approximately equal numbers of male and female samples. This prevents potential sex biases from influencing model performance. For multi-diagnostic datasets like HBN, we created multiple binary classification tasks (e.g., MDD vs. NC, ADHD vs. NC), each treated independently within a unified evaluation framework. By incorporating such a comprehensive and diverse benchmark, we are able to thoroughly assess the robustness, transferability, and clinical relevance of our proposed BrainGFM.

# O  Data Acquisition and Preprocessing

For all cohorts, resting-state fMRI data were collected with varying protocols and scanner parameters specific to each study site. All available resting-state fMRI data were preprocessed using the well-established fMRIPrep pipelineEsteban et al. [2019]. The T1-weighted image was corrected for intensity non-uniformity and then stripped skull. Spatial normalization was done through nonlinear registration, with the T1w referenceAvants et al. [2008]. Using FSL, brain features such as cerebrospinal fluid, white matter, and grey matter were segmented from the reference, brain-extracted

Table 13: Overview of 25 Brain Disorders Across Public Neuroimaging Datasets. We select balanced samples (e.g., HBN, ADNI) for downstream classification. Note that all downstream tasks have balanced numbers of male and female samples.

| Dataset | Disease/Disorder/Disability | Downstream Task | Age | Sample Size |
|---|---|---|---|---|
| ADHD200 | Att-ention-Deficit/Hyperactivity Disorder (ADHD) | ADHD vs. NC | 8-26 | 402/580 |
| ABIDE II | Autistic Spectrum Disorders (ASD) | ASD vs. NC | 5-64 | 581/733 |
| ADNI 2 | Alzheimers Disease (AD) | AD vs. NC | 55-89 | 91/100 |
| | Mild Cognitive Impairment (MCI) | MCI vs. NC | 55-89 | 168/200 |
| OASIS3 | Dementia (DM) | DM vs. NC | 45-88 | 290/300 |
| HBN | Major Depression Disorder (MDD) | MDD vs. NC | 6-20 | 261/310 |
| | Anxiety (ANX) | ANX vs. NC | 6-20 | 224/250 |
| | Oppositional Defiant Disorder (ODD) | ODD vs. NC | 6-20 | 519/319 |
| | Obsessive-Compulsive Disorder (OCD) | OCD vs. NC | 6-20 | 130/150 |
| | Language Disorder (LD) | LD vs. NC | 6-20 | 464/319 |
| | Specific Learning Disorder (SLD) | SLD vs. NC | 6-20 | 335/319 |
| | Enuresis (ENU) | ENU vs. NC | 6-20 | 279/319 |
| | Intellectual Disability (ID) | ID vs. NC | 6-20 | 129/150 |
| | Post-Traumatic Stress Disorder (PTSD) | PTSD vs. NC | 6-20 | 39/40 |
| | Encopresis (ECP) | ECP vs. NC | 6-20 | 68/70 |
| | Dysthymia (PDD) | PDD vs. NC | 6-20 | 85/90 |
| | Tourette Sydnrome (PS) | PS vs. NC | 6-20 | 68/70 |
| | Adjustment Disorder (AJD) | AJD vs. NC | 6-20 | 96/100 |
| | Provisional Tic Disorder (PTD) | PTD vs. NC | 6-20 | 68/70 |
| | Motor Disorder (MD) | MD vs. NC | 6-20 | 123/130 |
| | Speech Sound Disorder (SSD) | SSD vs. NC | 6-20 | 98/100 |
| | Communication Disorder (CD) | CD vs. NC | 6-20 | 43/45 |
| SubMex_CUD | Cocaine Use Disorder (CUD) | CUD vs. NC | 18-45 | 72/63 |
| UCLA_CNP | Schizophrenia (SCHZ) | SCHZ vs. NC | 21-50 | 50/55 |
| | Bipolar Disorder (BP) | BP vs. NC | 21-50 | 49/55 |
| REST-META-MDD | Major Depression Disorder (MDD) | MDD vs. NC | 21-50 | 1,252/1,101 |

T1 weighted imageZhang et al. [2000]. The fieldmap information was used to correct distortion in low-frequency and high-frequency components of fieldmap. Then, a corrected echo-planar imaging reference was obtained from a more accurate co-registration with the anatomical reference. The blood-oxygenation-level-dependent (BOLD) reference was then transformed to the T1-weighted image with a boundary-based registration method, configured with nine degrees of freedom to account for distortion remaining in the BOLD referenceGreve and Fischl [2009]. Head-motion parameters (rotation and translation parameters of volume-to-reference transform matrices) were estimated with MCFLIRT (FSL). BOLD signals were slice-time corrected and resampled onto the participant's original space with head-motion correction, susceptibility distortion's correction, and then resampled into standard space, generating a preprocessed BOLD run in MNI152NLin2009cAsym space. Automatic removal of motion artifacts using independent component analysis (ICA-AROMA)Pruim et al. [2015] was performed on the preprocessed BOLD time-series on MNI space after removal of non-steady-state volumes and spatial smoothing with an isotropic Gaussian kernel of 6 mm FWHM (full-width half-maximum).

# P  Definition and Description of Used Atlases/Parcellations

Table 15 provides a systematic comparison of eight widely used brain atlases and parcellations adopted in our study, including the number of parcels, construction type (functional or anatomical), year of release, and key design features. These atlases represent the most commonly applied frameworks in both functional and structural neuroimaging studies and are constructed based on diverse methodological principles, making them suitable for different modeling objectives in neuroscience research. Specifically, the Schaefer atlas series (Schaefer100/200/300) is derived from resting-state fMRI data using gradient-weighted clustering to generate spatially contiguous functional parcels. Each parcel is assigned to one of the Yeo 7 or 17 functional networks, preserving hierarchical organization and functional homogeneity. This design makes Schaefer atlases particularly suitable for

Table 14: Disorders in Different Categories and Their Datasets.

| Category | Disease/Disorder | Dataset(s) |
|---|---|---|
| **Neurodevelopmental Disorders** | Attention-Deficit/Hyperactivity Disorder (ADHD) | ADHD200 |
| | Autism Spectrum Disorder (ASD) | ABIDE II |
| | Language Disorder (LD) | HBN |
| | Specific Learning Disorder (SLD) | HBN |
| | Intellectual Disability (ID) | HBN |
| | Speech Sound Disorder (SSD) | HBN |
| | Communication Disorder (CD) | HBN |
| **Neurodegenerative Disorders** | Alzheimer's Disease (AD) | ADNI 2 |
| | Mild Cognitive Impairment (MCI) | ADNI 2 |
| | Dementia (DM) | OASIS3 |
| **Mood and Anxiety Disorders** | Major Depression Disorder (MDD) | HBN, REST-META-MDD |
| | Anxiety (ANX) | HBN |
| | Post-Traumatic Stress Disorder (PTSD) | HBN |
| | Adjustment Disorder (AJD) | HBN |
| | Mild Depression Disorder (PDD) | HBN |
| | Bipolar Disorder (BP) | UCLA_CNP |
| **Obsessive-Compulsive and Impulse Control Disorders** | Obsessive-Compulsive Disorder (OCD) | HBN |
| | Oppositional Defiant Disorder (ODD) | HBN |
| | Intermittent Explosive Disorder (IED) | HBN |
| **Motor Disorders** | Tourette Syndrome (TS) | HBN |
| | Motor Disorder (MD) | HBN |
| | Provisional Tic Disorder (PTD) | HBN |
| **Substance Use Disorders** | Cocaine Use Disorder (CUD) | SubMex_CUD |
| **Psychotic Disorders** | Schizophrenia (SCHZ) | UCLA_CNP |

Table 15: Comparison of Common Brain Atlases and Parcellations Used in Our Study.

| Atlas/Parcellation | Parcel Num | Type | Year | Key Features |
|---|---|---|---|---|
| **Schaefer100** | 100 | Functional | 2018 | Based on resting-state fMRI; each parcel belongs to Yeo 7/17 networks; spatially contiguous; gradient-weighted clustering. |
| **Schaefer200** | 200 | Functional | 2018 | Higher resolution; suitable for fine-grained functional connectivity or graph modeling. |
| **Schaefer300** | 300 | Functional | 2018 | Even finer granularity; suitable for detailed graph analysis but may increase noise. |
| **Shen268** | 268 | Functional | 2013 | Group-wise ICA-based; spatially contiguous; widely used in functional connectomics and GNNs. |
| **Power264** | 264 | Functional | 2011 | Functional hubs as spheres; not spatially contiguous; commonly used in network neuroscience. |
| **Gordon333** | 333 | Functional | 2016 | Combines local gradient and network assignment; fine resolution. |
| **AAL116** | 116 | Anatomical | 2002 | Based on anatomical landmarks; widely used in structural/functional neuroimaging; standard in SPM. |
| **AAL3v1** | 170+ | Anatomical | 2020 | Updated AAL; includes more detailed subcortical and cerebellar regions. |

functional connectivity analysis and graph neural network modeling. The availability of multiple spatial resolutions enables systematic evaluation of model behavior under coarse- to fine-grained parcellations. The Shen268 atlas, constructed via group-level independent component analysis (ICA), offers spatially contiguous and inter-subject consistent functional parcels and has become a standard in GNN-based fMRI research. In contrast, the Power264 atlas identifies spherical regions centered on functional hubs without enforcing spatial continuity. Although less anatomically constrained, it is widely used in network neuroscience, particularly for studying nodal centrality and modular organization. The Gordon333 atlas integrates local gradient information and functional network

assignment to define high-resolution, functionally coherent brain regions, enabling precise modeling of functional boundaries. In terms of anatomical atlases, the AAL116 atlas is one of the earliest structural templates, based on anatomical landmarks and extensively used in both structural and functional neuroimaging studies. It remains the default parcellation in tools such as SPM. The AAL3v1 atlas is an updated version of AAL116, providing finer subdivisions of subcortical and cerebellar regions for enhanced spatial coverage and granularity, supporting more detailed structural-functional integration.

By incorporating both functional and anatomical atlases, as well as a wide range of spatial granularities (from 100 to 333 parcels), our study is designed to comprehensively evaluate the adaptability, scalability, and generalization capacity of brain graph models across heterogeneous parcellation strategies. This diverse atlas configuration facilitates pre-training under varied topological priors and enables robust transfer to downstream tasks involving unseen atlases or disorders. Such design is critical for building generalizable BrainGFMs capable of adapting to diverse neuroimaging datasets and real-world clinical scenarios.

# Q   Broader Impact

Our proposed Brain Graph Foundation Model (BrainGFM) is designed to be a unified and versatile architecture for graph-based modeling of brain data. While our current experiments focus on resting-state functional MRI (rs-fMRI), the model is modality-agnostic and readily extensible to other neuroimaging modalities, including task-based fMRI (task-fMRI), electroencephalography (EEG), diffusion tensor imaging (DTI), and magnetoencephalography (MEG). These diverse modalities can be represented as brain graphs, constructed from temporal correlations, structural connectivity, or stimulus-evoked activity patterns, making BrainGFM a generalizable framework for multi-modal neuroscience applications.

The ability to transfer knowledge across data types, brain atlases, and clinical conditions enables BrainGFM to benefit a wide range of downstream tasks, including biomarker discovery, mental disorder diagnosis, and brain-computer interface (BCI) development. Its strong pre-training on large-scale brain graphs makes it particularly valuable for low-resource or small-sample settings.

# R   Limitation and Future Works

While our work successfully constructed a large-scale fMRI dataset for pre-training, certain limitations remain. Due to the significant manual effort involved, we were unable to include all datasets from the OpenNeuro platform Markiewicz et al. [2021], particularly the large number of task-based (non-resting-state) fMRI datasets. In addition, because of financial constraints, we were not able to incorporate fMRI data from the UK Biobank Bycroft et al. [2018], including both resting-state and task-based scans, as access to this dataset requires paid licensing and ongoing maintenance costs.

In future work, our dataset can be further expanded by incorporating additional resources such as the full OpenNeuro repository and the UK Biobank dataset. This would enable the construction of an even larger pre-training corpus for BrainGFMs. Moreover, combining task-based and resting-state fMRI data could lead to a more comprehensive representation of brain dynamics. We believe that with the inclusion of more diverse datasets and task-based fMRI, the performance and generalization ability of BrainGFM can be further enhanced.

