# OpenReview forum: "A Brain Graph Foundation Model: Pre-Training and Prompt-Tuning for Any Atlas and Disorder"
_NeurIPS.cc/2025/Conference — Submitted to NeurIPS 2025_

### Official Review · Reviewer_CS9f · 2025-06-14

**Clarity:** 1
**Significance:** 2
**Originality:** 2
**Rating:** 3
**Confidence:** 4

**Summary:**

This paper introduces the Brain Graph Foundation Model (BrainGFM), a novel graph-based framework for analyzing fMRI data, pretrained and tested on extensive and diverse datasets parcellated by different atlas. BrainGFM leverages a pre-training paradigm that combines graph contrastive learning with graph masked autoencoders to effectively model brain connectivity. The model incorporates graph prompts optimized via meta-learning for few-shot generalization to unseen disorders, and language prompts that provide semantic guidance for zero-shot transfer to new tasks, atlases, and disorders. BrainGFM has achieved superior performance and efficiency compared to previous brain foundation models.

**Questions:**

Please refer to the weakness section.

**Ethical Concerns:**

["NO or VERY MINOR ethics concerns only"]

**Final Justification:**

The discussion on previous work should be clearer now, with addtional results compared to Brain-JEPA added. However, the performance gain seems incremental, sometimes even worse compared to SOTA. And the paper has made a very strong claim that it can be adapted to any atlas or disorder, which requires more evaluation and validation.

**Limitations:**

yes

**Quality:**

2

**Strengths And Weaknesses:**

Strengths:

Incorprating heterogeneous atlas in one single model is a novel setting, which is also quite useful in fMRI domain.

The datasets and downstream tasks span quite diverse brain disorders.

The model has demonstrated zero-shot capability, which should be the first time for brain foundation models.

The model works better than previous SOTA models on brain graph like BrainMass.

Weakness:

What is the difference between "time-series-based" models and "ROI-based" models mentioned in intro? This is confusing, for example, BrainLM [1] and Brain-JEPA [2] both have the parcellated time series as input.

“While ROI-based brain FMs are more efficient, they often neglect inter-regional connectivity” What does this mean? The whole point about functional connectivity is to capture inter-regional connectivity?

The authors may want to say time series based and connectome based? The ROI based is confusing as those time series were parcellated as well.

The discussion on those previous works was really confusing. In addition, since many of the previous brain foundation models were based on time series,  it would be beneficial to more clearly articulate the benefits of FC as input in this domain from the neuroscience perspective.

The lack of formatting distinction between main content and citations (e.g., xxx et al.) impairs readability.

There are previous works on prompt tuning of brain models including [3] and [4] which were missed in the related work. The authors should also discuss Meta-Matching [5] which represents a widely used meta learning framework in FC.

Why 100-300 ROIs were chosen in Schaefer atlas? How about the 400 and more? How is the age distribution like in pretraining datasets? It seems that the majority of the pretraining data is childhood and young adult, with the middle age to aging population under-represented.

Brain gradient positioning [2] has been proved to work well for fMRI data, which should be compared in positional encoding part as well.

line 211-213, using limited samples to fine tune large models, it leads to overfitting instead of "underfitting"?

For the meta learning, does it mean that data from multiple diseases is required during this stage? which may be a limitation in real-world application?

Brain-JEPA was missed in comparison, which should be a more advanced model than BrainLM.

There are significant concerns about the reliability of their checklist responses and the reproducibility of this work. In checklist Question 5, the authors state that "anonymized code and benchmark instructions will be released in the supplementary material," yet no supplementary material is provided. On the other hand, while the authors claim to include error bars for their results, Table 1 shows no error bars for the main experimental findings.

[1] BrainLM: A foundation model for brain activity recordings. ICLR 2024.

[2] Brain-JEPA: Brain Dynamics Foundation Model with Gradient Positioning and Spatiotemporal Masking. NeurIPS 2024.

[3] BrainPrompt: Multi-Level Brain Prompt Enhancement for Neurological Condition Identification. MICCAI 2025.

[4] Prompt Your Brain: Scaffold Prompt Tuning for Efficient Adaptation of fMRI Pre-trained Model. MICCAI 2024.

[5] Meta-matching as a simple framework to translate phenotypic predictive models from big to small data. Nature Neuroscience 2022.

---

> ### Author Rebuttal · Authors · 2025-07-30
>
> ### 🟩 **Clarifying Model Categorization: Time-Series vs. Connectome vs. Graph**
>
> Thank you for pointing out the ambiguity in our terminology. We agree that “connectome-based” is a more accurate term than “ROI-based,” as both categories utilize parcellated ROI-level time series. We have revised the manuscript to consistently use “connectome-based” to avoid confusion and better reflect the underlying data representation.
>
> To address this concern, we have significantly clarified the discussion of prior brain foundation models by categorizing them based on their input modality and modeling strategy. Specifically, we have updated Supplementary Table 13 by renaming the column from *Model Type* to *Input Type* and replaced “ROI-based” with the more precise term *Connectome-based*. The revised categorization is as follows:
>
> 1. **Time-series-based models**: e.g., BrainLM, Brain-JEPA. These models take the parcellated ROI-level fMRI time series as input (typically of shape ROI × time) and model temporal dependencies using sequence-based architectures such as Transformers. While expressive, they are computationally intensive due to high input dimensionality and memory demands.
>
> 2. **Connectome-based or FC-based models** (formerly “ROI-based”): e.g., BrainMass, BrainNPT. These models operate on precomputed static functional connectivity (FC) matrices derived from time series (typically ROI × ROI), offering efficiency and interpretability but limited dynamic modeling.
>
> 3. **Graph-based models (ours)**: We build a brain graph where ROIs are nodes, node features are global connectivity profiles (i.e., rows from the FC matrix), and edges define sparse topologies via functional/structural priors (e.g., top-k or k-NN). This enables localized message passing and self-supervised objectives tailored for brain graphs, capturing both global and relational structure at low cost.
>
> ---------------------------
>
> ### 🟦 **On Inter-regional Connectivity and Graph Structure**
>
> While it is true that our node features are derived from each ROI’s Pearson correlation with all other ROIs (i.e., a row of the FC matrix), we emphasize that the **explicit graph structure (edges)** plays a distinct and complementary role in our model.
>
> - The **node features** capture global functional profiles.
> - The **adjacency matrix** (top-k, k-NN) encodes a structural prior, restricting message passing to sparse, biologically relevant neighbors.
>
> Although inter-regional relationships are implicit in the node features, the graph topology:
> - Encourages **localized and interpretable propagation**, and
> - **Improves generalization** by enforcing biologically informed constraints.
>
> This is fundamentally different from flat FC-matrix inputs used in MLPs or CNNs, which only capture global connectivity patterns without explicit structural modeling. In contrast, our graph-based approach combines global functional profiles (from FC-derived node features) with local relational structure (via sparse graph edges), enabling both global and local information propagation through message passing over the graph topology. This dual modeling enhances higher-order representation learning and improves generalization.
>
> -------------------------------
>
> ### 🟩  **Changing ROI to Connectome/FC**
>
> We have revised the terminology of ROI throughout the paper to use Connectome/FC-based, which more accurately reflects the nature of the input.
>
> ------------------------
>
> ### 🟥 **Clarifying Benefits of Functional Connectivity as Input**
>
> We thank the reviewer for raising the neuroscience perspective. Functional connectivity (FC) matrices—representing ROI-wise correlations—are a widely adopted representation in neuroscience for capturing stable inter-regional dependencies.
>
> Compared to raw time series, FC-based inputs:
> 	•	Are computationally efficient
> 	•	Offer interpretable and stable representations
> 	•	Are robust to frame-level noise
>
> However, this efficiency comes at the cost of temporal resolution—FC representations inherently lack the ability to model fine-grained spatiotemporal dynamics. In contrast, time-series-based methods can capture both local temporal patterns and global dependencies, but they are computationally expensive due to operating over full-length ROI × T sequences.
>
> Our proposed graph-based model builds upon the efficiency of FC by treating each node’s feature as its FC profile, and introduces an explicit graph structure, which enables not only the modeling of global connectivity patterns, but also the extraction of local relational features through message passing on the graph edges using GCN. Thus, our method strikes a balance between efficiency, interpretability, and expressiveness, making it well-suited for downstream clinical tasks such as diagnosis.
>
> --------------------
>
> ### 🟪 **Formatting and Citations**
>
> We have revised all inline citations to follow the standard format (e.g., *Thomas et al., 2022*) and ensured clear visual distinction between citation text and narrative content for improved readability.
>
> ---------------
>
> ### 🟩 **Missing Related Works: Prompt Tuning and Meta-Matching**
>
> We appreciate the suggestion to include missing related works.
>
> - We have added discussions of prior **prompt tuning approaches** for brain models ([3], [4]) in the Related Work section. These works provide valuable baselines and motivation for prompt-driven adaptation in neuroimaging.
> - We now explicitly compare our approach with **Meta-Matching** ([5]) in Supplementary Section I.
>
> While both aim to improve generalization under limited labels:
> - **Meta-Matching** performs **post-hoc alignment** via a linear mapping without episodic training.
> - Our method performs **full episodic meta-learning**, involving inner-loop adaptation and outer-loop meta-optimization.
>
> This enables broader and stronger transfer across diverse tasks, including **zero-shot generalization to unseen disorder-atlas combinations**.
>
> -------------------------------
>
> ### 🟦 **Choice of 100–300 ROI Schaefer Parcellations**
>
> The **100–300 ROI resolutions** in the Schaefer atlas provide a strong trade-off between **granularity and computational cost**. While higher-resolution atlases (e.g., 400+) offer more detailed spatial information, they introduce substantial memory and runtime burdens, especially in large-scale pretraining. Our choice balances anatomical fidelity with scalability and aligns with prior works.
>
> ----------------------------------------
>
> ### 🟥 **Age Distribution of Pretraining Cohorts**
>
> We acknowledge the concern about age distribution. While our dataset is skewed toward younger populations due to inclusion of **ABCD, HBN**, etc., we also include datasets like **OASIS, ADNI, HCP-Aging, and HABS**, which cover older adults. In future work, we plan to incorporate large-scale aging cohorts such as **UK Biobank** to enhance age diversity. We have added a discussion of this limitation in the revised manuscript.
>
> ---------------------------
>
> ### 🟪 **Positional Encoding: Brain Gradient vs. RWSE**
>
> We have included **Brain Gradient Positioning (BGP)** in the ablation study of positional encodings. The results show that **BGP and RWSE** achieve comparable performance when applied to graph transformers in brain modeling, confirming that Brain Gradient PE is a strong and biologically meaningful encoding for brain networks, indicating both effectively capture brain topological priors.
>
>
> | **[PE] Type**  | **Pre-Trained** | **Efficiency** | **Performance (ACC / AUC)** |
> |--------------|--------------|-------------|------------|
> | w/o [PE]     | ✗  | Very Fast    | 65.2 / 67.1   |
> | w/o [PE]    | ✓    | Very Fast   | 69.5 / 71.7 |
> | Laplacian [PE]     | ✓  | Slow  | 69.2 / 71.3 |
> | Node Degree [PE]       | ✓  | Fast | 68.4 / 70.5 |
> | Brain Gradient [PE]    | ✓    | Fast      | 70.4 / 72.5     |
> | RWSE [PE]              | ✗    | Fast    | 66.1 / 68.0          |
> | **RWSE [PE]**          | **✓**    | **Fast**       | **70.5 / 73.3**     |
>
> While BGP performs well, our graph-based model benefits slightly more from graph-specific encodings like RWSE. Nonetheless, Brain Gradient PE remains highly promising and could be further enhanced by adapting it into a graph-aware formulation.
>
> ------------------
>
> ### 🟩 **Typo Correction: Overfitting vs. Underfitting**
>
> Thank you for catching the error in line 211–213. You are absolutely right—the correct term is **overfitting**, not underfitting. We have corrected this in the updated manuscript.
>
> ------------------
>
> ### 🟦 **Meta-Learning with Multi-Disease Data: Practicality**
>
> Our meta-learning framework is flexible:
>
> - If **only one disease** is available, we can construct pseudo-tasks across **sites/sessions/subjects** for within-disorder meta-learning.
> - If **multiple diseases** are available (even with few samples), we benefit from task-level transfer, significantly reducing per-disease annotation needs.
>
> Thus, while multi-disease meta-learning enhances generalization, the method remains practical in **single-disorder** or **few-label** regimes.
>
> ----------------------
>
> ### 🟥 **Brain-JEPA Baseline Inclusion**
>
> Thank you for the reminder. We have added **Brain-JEPA** as a baseline in Table 1 (main paper) and Table 13 (supplementary). Brain-JEPA is a more advanced model than BrainLM and shows **comparable performance** to BrainGFM, validating the strength of high-capacity time-series pretraining.
>
> We fully acknowledge that Brain-JEPA is a strong and representative method of the time-series-based paradigm. Its inclusion highlights the advantages of modeling fine-grained temporal dynamics directly from parcellated ROI time series.
>
> --------------------
>
> ### 🟪 **Reproducibility: Code and Error Bars**
>
> - We have now uploaded the anonymized code to an anonymous repository in the revised version.
> - We have updated **Table 1** to include **error bars (standard deviations)** computed from multiple runs.

---

> > ### Comment · Reviewer_CS9f · 2025-08-02
> >
> > Thank the authors for their response.
> >
> > While the response regarding some clarifications address part of the concerns, those ones corresponding to additional results/consolidation are actually invalid. For example, "We now explicitly compare our approach with Meta-Matching ([5]) in Supplementary Section I.", " We have added Brain-JEPA as a baseline in Table 1 (main paper) and Table 13 (supplementary).", "We have now uploaded the anonymized code to an anonymous repository in the revised version.", and "We have updated Table 1 to include error bars (standard deviations) computed from multiple runs." Such response does not help, as the reviewer can not see any "actual" results. The only way is to put the tables in the rebuttal as we can not see your revised version.
> >
> > In addition to the some missing tables, some discussions are also missed at this stage, for example, "We have added discussions of prior prompt tuning approaches for brain models ([3], [4]) in the Related Work section." How to evaluate the discussion given only such comment?
> >
> > Regarding "Choice of 100–300 ROI Schaefer Parcellations", it is true that less ROI lead to better efficiency, but how does the model perform given finer parcellation? Is there any scaling we expect to see? The authors have made a very strong claim, like in title: "for **ANY** atlas". The evaluation in the current form can not support such strong claim.

---

> ### Author Response · Authors · 2025-08-03
>
> ### 🟪 **Clarification on Additional Results and PDF Limitations**
>
> We thank the reviewer for the detailed review and constructive feedback.
> Due to the word limit and the restriction on uploading updated PDFs, we were unable to include all results in the original submission.
> We now present them here as additional evidence for further clarification.
>
> ---
>
> ### 🟩 **Comparison with Meta-Matching in Supplementary Section**
>
> We added a comparison with **Meta-Matching**.
> We summarize the key differences in the table below:
>
> | **Aspect**| **Meta-Matching**| **Our Meta-Learning Framework**|
> |---|---|----|
> | **Meta-learning Paradigm**| No explicit meta-learning; post-hoc linear adaptation | Prompt-based meta-learning with inner/outer optimization |
> | **Adaptation Mechanism**| Learns output mapping $g(f_{\text{pre}}(x))$ via linear regressor| Learns task-specific prompt tokens injected into the backbone model|
> | **Gradient Flow**| Only regressor is trained; backbone remains fixed | Prompts are updated via inner-loop gradients; backbone frozen |
> | **Task Definition**| Each small dataset = separate task; no shared structure | Tasks defined by (disorder, atlas) tuples sampled from unified distribution|
> | **Update Scope** | No model component is updated during adaptation | Only prompt modules are adapted; base encoder remains fixed |
>
> ---
>
> ### 🟦 **Clarification on Brain-JEPA Results and Code Availability**
>
> We have evaluated **Brain-JEPA** on all 10 datasets.
> Due to space limits, only results on **BrainMass**, **Brain-JEPA**, and **BrainGFM** are shown on 5 datasets, with **standard deviations** included.
>
> **Table: Performance Comparison (mean ± std) on 5 Datasets  (AUC/ACC/SEN/SPE)**
>
> | **Method** | **PT** | **ADHD200 (ADHD)**| **ABIDE II (ASD)** | **ADNI 2 (AD)** | **HBN (MDD)**| **HBN (ANX)**  |
> |----|---|---|---|---|---|----|
> | **BrainMass**| Yes| 67.0±2.3 / 67.5±2.0 / 64.7±2.8 / 71.1±1.8  | 68.9±2.1 / 70.1±1.9 / 69.5±2.4 / 66.3±2.6 | 77.8±2.3 / 82.7±2.7 / 72.6±2.9 / 81.4±2.5    | 76.9±2.4 / 80.2±2.8 / 83.6±2.5/ 76.1±2.6 | 81.0±2.7 / 84.0±2.2 / 80.3±2.3 / 81.9±2.9 |
> | **Brain-JEPA**| Yes| 69.8±1.9 / 71.6±2.0 / 66.2±1.6 / 72.9±2.4| 70.1±1.7 / 73.8±1.8 / 70.1±1.9 / 69.3±1.7| 79.1±2.2 / 84.3±1.8 / 76.8±2.2 / 83.6±2.1 | 83.4±1.7 / 85.9±1.3/ 84.3±1.9 / 77.5±1.5 | 85.4±1.8 / 86.7±2.0 / 85.3±2.2 / 83.7±1.9  |
> | **BrainGFM** | Yes| 70.6±1.6 / 72.2±1.5 / 67.3±1.7 / 73.4±2.1| 71.2±1.9 / 73.5±1.4 / 70.4±1.5 / 69.8±1.7 | 80.3±2.6 / 85.1±2.2 / 76.2±1.5 / 84.4±1.7 | 83.6±1.6 / 85.5±1.7 / 85.8±1.6 / 77.9±1.9| 85.2±1.6 / 86.3±2.1 / 87.7±1.9 / 82.6±1.7 |
>
> The code has been uploaded to an anonymous repository, and we will release it upon paper acceptance.
> As per rebuttal guidelines, we are*not allowed to share links, even to anonymous repositories.
>
> ---
>
> ### 🟥 **Clarification on Prompt Tuning Discussion**
>
> To reflect recent work, we added to Section 2.2:
>
> > *"Recent studies have also introduced brain prompt-tuning methods **BrainPrompt and ScaffoldPrompt** to adapt pre-trained fMRI models."*
>
> In the Appendix, we clarify the distinction:
>
> > *"While prior works like BrainPrompt and Scaffold Prompt adapt fMRI models using token- or scaffold-style prompts on time-series or ROI-level features, our method introduces a graph-structured prompt designed specifically for brain graphs. These prompts include learnable node and edge parameters aligned with brain graph topology, enabling better structural encoding. Unlike prior methods that tune prompts per task, we adopt a meta-learning approach to generalize across multiple disorder-atlas pairs, supporting few-shot and zero-shot adaptation through a unified and scalable prompt mechanism.”*
>
> ---
>
> ### 🟧 **Clarification on 100–300 ROI Schaefer Parcellations**
>
> For each individual disorder, a finer parcellation (i.e., more ROIs) generally results in better performance.
> However, the optimal parcellation scale is not universal—it varies across disorders.
>
> ---
>
> ### 🟩 **Clarification on “ANY Atlas” Claim**
>
> To support the claim, we conducted:
>
> 1. **Cross-Atlas Transfer**: Train on one atlas (e.g., Schaefer100), evaluate on another (e.g., AAL, Power).
> 2. **Cross-Parcellation Transfer**: Train on one scale, evaluate on finer/coarser scales.
>
> **Table: Ablation experiments on the ADNI2.  The reported improvements (%) indicate the performance gain over models without pre-training.**
>
> | **Type** | **Pre-training** | **Fine-tuning** | **Performance**|
> |---|---|---|---|
> | **Cross-Atlas**| Schaefer100| AAL116| 3.3% ±1.5% ↑|
> |  | Schaefer100|Power264| 2.7% ±1.3% ↑|
> |  | Schaefer200 + AAL116| Schaefer100| 3.2% ±1.6% ↑|
> | **Cross-Parcellation**| Schaefer100| Schaefer300| 2.7% ±1.4% ↑|
> |  | Schaefer100 + Schaefer300| Schaefer200| 3.5% ±1.5% ↑ |
> |  | AAL116| AAL3v1|3.1% ±1.3% ↑|
>
> ---

---

> > ### Comment · Reviewer_CS9f · 2025-08-04
> >
> > Thank the authors for adding these results explicitly.
> >
> > I have increased the score to acknowledge the clarification and addtional results compared to Brain-JEPA and brain gradient positioning. While I still think the central claim of the model flexibility is too strong which requires further evaluation, and the performance compared to previous state-of-the-art models seems incremental.

---

> ### Author Response · Authors · 2025-08-04
>
> ### 🟨 **Response to Concern on "Any Atlas and Disorder" Claim**
>
> We appreciate the reviewer’s feedback and acknowledge that the phrase “for Any Atlas and Disorder” in our original title may have been too strong.
> While our model is designed to adapt flexibly across a wide range of brain atlases (e.g., Schaefer100–300, AAL, Power) and neurological disorders (e.g., MDD, ASD, AD), we agree that the word "**any**" may overstate the current empirical coverage.
>
> To address this, we have revised the title to use the term "**broad**" instead of "**any**", reflecting a more measured and accurate description of our model’s generalization capacity.
>
> So the new title will be "A Brain Graph Foundation Model: Pre-Training and Prompt-Tuning across Broad Atlases and Disorders".
>
> We hope this clarifies our intent and aligns better with the current evaluation scope.
>
> ---
>
> ### 🟧 **Clarification on Full-Shot Performance Compared to Brain-JEPA**
>
> We appreciate the recognition of our clarification and additional results.
> We agree that **Brain-JEPA** achieves strong performance, particularly in full-shot settings.
> However, our model is not solely focused on full-shot accuracy, but rather on introducing a new foundation model paradigm for brain graphs that is:
>
> 1. More memory-efficient and computationally lighter than time-series-based models like Brain-JEPA. Our graph-based architecture enables training and inference on large-scale fMRI cohorts with **significantly reduced memory usage**.
> 2. Specifically designed to support **few-shot and zero-shot generalization**, which is crucial for real-world clinical settings where labeled data is scarce.
> 3. Aligned with the emerging trend of foundation models, our work is the first to explore graph-based foundation pre-training for brain connectomics.
>
> Thus, the **core value of our work lies not only in performance**, but also in proposing a new, practical and generalizable modeling direction that balances scalability, efficiency, and adaptability.
>
> We hope the reviewer finds this clarification helpful.

---

### Official Review · Reviewer_4mpv · 2025-06-16

**Clarity:** 3
**Significance:** 3
**Originality:** 3
**Rating:** 5
**Confidence:** 3

**Summary:**

The manuscript is well written and presents an in-depth exploration of integrating multiple brain parcellation schemes within a unified framework for foundation models in Graph Theory. This study addresses the challenges of few-shot and zero-shot learning in neuroimaging. The authors introduce BrainGFM, a novel model that is pre-trained to use a combination of graph-specific strategies, incorporating multiple parcellations to boost performances.

This work uses data from 27 widely used fMRI datasets from diverse sites, and 25 common neurological and psychiatric disorders. This scale of data enhances the model’s robustness and generalizability.

**Questions:**

1) While the manuscript makes a strong case for leveraging large-scale data to address several challenges, it appears to underemphasize critical aspects of data harmonization prior to processing, as well as class imbalances, both within and across datasets. It remains unclear whether variables such as age and sex distributions are uniform across cohorts, or whether control subjects are consistently included in each dataset. These factors can significantly influence model performance and generalizability. I also wonder whether there is any correlation between the model’s overall accuracy and the balance of these demographic and clinical variables. While I appreciate that the primary focus is on advancing the technical aspects of the foundation model, the curation and construction of the input data are equally vital and warrant more thorough discussion.

2) In Table, I would be curious to have more context about the conditions and the datasets, for these conditions I would like to know more about the clinical significance. In some case sensitivity is a useless metric and specificity is much more important (AD for example, a clinician knows something is wrong, but being certain about a diagnostic is much harder), and what would be the target of clinical significance ? In my experience specificity is by far the most important one. And overall for this condition what is the overall target for the field, is it already achieved with a Vanilla GCN (probably not), but are we getting closer with the proposed method? This information would be very important context, in a clinical setting what is the sensitivity/specificity expected for a visit with a neurologist? Or is the goal to screen people before they see a neurologist (and in that case what is the expected values for sensitivity/specificity?)

**Ethical Concerns:**

["NO or VERY MINOR ethics concerns only"]

**Final Justification:**

I believe the author adressed the limitations I observed as well as the other reviewers feedback. Considering my initial review was 'optimistic', I think my initial rating should not change, but the changes to the manuscript and clarifications are in line with what I was expecting.

**Limitations:**

yes

**Paper Formatting Concerns:**

Figure 6, (a) Constrcution

Table 1: the second set seems to be using a darker gray for the (ours) row, making hard to see which one is gray vs pink.

Line 64: Pre-trained brain FMs needs (remove s) to be fine-tuned to various downstream tasks
(FMs being plural I think)

Line 131: adaption -> adaptation?

Line 263: The pre-trained FMs significantly outperforms (no s)

**Quality:**

4

**Strengths And Weaknesses:**

The manuscript is very clear, I particularly enjoy the Methodology section where each contribution is justified with a clear and direct motivation. The manuscript is very well written and well organized. The experiment is adequate for the presented hypotheses.
The 6 subsections to the experiments (comparison with references, ablation Full/Few/Zero-Shot, parcellations, time series/ROI/Graph FM, ablation parcellation, ablation on FM pre-training) are well designed to justify and quantify the author's contributions.

One weakness (which is not that problematic) is that the authors are pushing multiple high complexity 'story' into a somewhat short manuscript with a lot of very important details in supplementary materials. The contributions are such a high level, with so many layers of expected knowledge that the clarity of the main text is decreasing and the impact potentially obscure by the complexity of the multidimensional experimentations. There is no clear solution, I just wanted to share this.

---

> ### Author Rebuttal · Authors · 2025-07-30
>
> ### 🟩 **Clarity and Complexity – Integration of Multiple Contributions**
>
> We sincerely thank the reviewer for the thoughtful feedback and recognition. We understand the concern regarding the complexity and density of our manuscript. This is primarily because our work presents a comprehensive and integrated framework that unifies both pre-training and downstream fine-tuning, covering not only standard settings but also few-shot and zero-shot scenarios. Rather than addressing each component in isolation, we aim to demonstrate how they collectively contribute to building a Brain Graph Foundation Model — a novel and unified perspective that, to the best of our knowledge, is the first attempt in this direction.
>
> Notably, prior brain foundation models have largely focused on pre-training alone, often overlooking the critical yet practically important downstream tuning strategies, especially in few-shot and zero-shot settings. However, these settings are highly relevant in real-world clinical applications, where labeled data is often scarce or entirely unavailable for new disorders or sites. Our work deliberately addresses this gap by integrating a full-stack solution encompassing pre-training, prompt tuning, meta-learning, and generalization to unseen tasks. As a result, the manuscript may appear information-dense, but we believe this comprehensiveness is essential to convey the full scope and practical utility of our approach.
>
> The extensive experimental design and multilayered methodology reflect our commitment to presenting a complete and practical solution, rather than a partial or fragmented one. Given the scope and ambition of our work, we have carefully organized the supplementary materials to ensure that all technical and implementation details are accessible to interested readers without overwhelming the main text.
>
> -------------------------------------------------------------------------------
>
> ### 🟦 **Q1: Data Harmonization and Demographic Balance**
>
> Thank you for raising this important concern. We would like to clarify that one of the **core motivations and advantages** of developing a brain foundation model is precisely to address the challenge of **data heterogeneity across cohorts**. By pre-training on **large-scale and diverse datasets** that encompass a wide range of acquisition sites, protocols, and subject demographics, **BrainGFM is explicitly designed to learn representations that are robust to variations across datasets**. This pretraining paradigm naturally performs **data harmonization** by encouraging the model to generalize across heterogeneous input distributions, rather than requiring explicit harmonization steps prior to training. This is a key distinction from traditional models and a central benefit of the foundation model framework.
>
> In terms of **class imbalance**, especially for downstream tasks, we **carefully constructed balanced evaluation datasets** by ensuring **demographic (e.g., sex)** and **clinical label distributions** are as uniform as possible. Specifically, we followed the **sampling strategy adopted by BrainMass** to create downstream classification datasets with **balanced class labels** and **controlled gender ratios**. These efforts help to mitigate the potential **confounding effects** of demographic imbalance on model evaluation.
>
> -------------------------------------------------------------------------------
>
> ### 🟥 **Q2: Clinical Significance of Metrics and Use Scenarios**
>
> Thank you for this valuable and clinically grounded comment. We appreciate your emphasis on the importance of understanding each disorder’s diagnostic context, especially the relevance of **sensitivity versus specificity** in clinical decision-making. We address these points below by referring to the presented tables and the design of our evaluation pipeline.
>
> In our submission, we already provide detailed context for each brain disorder used in evaluation. Specifically, **Table 15** outlines the **target diagnosis**, **age range**, and **sample sizes** for each downstream classification task. The tasks span a diverse set of brain disorders across both **neurodevelopmental** (e.g., ADHD, ASD) and **neurodegenerative/psychiatric** conditions (e.g., AD, MDD, PTSD). For each task, we constructed **balanced datasets** in terms of both class distribution and gender (as described in Table 15 and Table 14), following the protocols of **BrainMass** and other prior works. This ensures that any performance differences are not **confounded by class imbalance or demographic skew**.
>
> Regarding the **clinical relevance of evaluation metrics**, we agree that the importance of **sensitivity vs. specificity** varies across tasks:
>
> - For **Alzheimer’s Disease (AD)**, specificity is particularly critical in clinical settings because clinicians often suspect cognitive decline, and a high-specificity model helps reduce **false positives** and **overdiagnosis**. In our results (Table 1), **BrainGFM achieves 84.4% specificity and 85.1% AUC on ADNI2**, significantly outperforming non-pretrained baselines (e.g., GCN 73.4% SPE, BrainLM 78.3% AUC but 81.4% SPE). This demonstrates that BrainGFM narrows the gap toward the **clinical diagnostic specificity target**.
>
> - For early screening tasks such as **ASD (ABIDE II)** or **ADHD (ADHD200)**, **high sensitivity** is clinically preferred to avoid missing early-stage cases. For ASD, BrainGFM achieves **70.4% sensitivity**, which is competitive given the task complexity and variability across ABIDE sites. For ADHD, we observe **67.3% sensitivity**, again higher than all baselines. These results suggest that our model aligns well with the clinical goal of **early detection**, where **missing true positives** can delay interventions.
>
> To reflect these differing priorities, we will update **Table 1** in the revision to **highlight (e.g., bold)** the **most clinically relevant metric for each task** (e.g., specificity for AD, sensitivity for ASD), and include a short summary in the supplementary text indicating the **expected clinical performance range for each disorder** (where available). This will help readers better interpret how close our model is to the **real-world utility level**.
>
> Finally, we clarify that **BrainGFM is designed to be flexible across both clinical and screening scenarios**:
>
> - For disorders like **AD or MDD**, we envision it as a **clinical decision support tool** used **after initial suspicion**, requiring **high specificity**.
> - For neurodevelopmental disorders such as **ASD and ADHD**, we see it as a **pre-clinical screening model**, prioritizing **sensitivity** to identify high-risk individuals early.
>
> This intended use distinction will also be **explicitly stated** in the revised manuscript.
>
> -------------------------------------------------------------------------------
>
> ### 🟪 **Paper Formatting Concerns**
>
> We thank the reviewer for the careful proofreading. We have **corrected all grammatical issues** and have thoroughly reviewed the entire manuscript to ensure that even **minor errors** have been addressed. We greatly appreciate your attention to detail, which helped us further improve the **quality and clarity** of the paper.
>
> -------------------------------------------------------------------------------

---

### Official Review · Reviewer_heFS · 2025-06-29

**Clarity:** 3
**Significance:** 3
**Originality:** 3
**Rating:** 4
**Confidence:** 3

**Summary:**

This paper introduces BrainGFM, a novel brain graph foundation model leveraging a graph-based pre-training paradigm for fMRI data. Key strengths include its ability to unify heterogeneous fMRI datasets, its robust performance across various brain atlases and disorders through innovative graph and language prompt-tuning, and its demonstrated superior efficiency and effectiveness compared to existing foundation models.

**Questions:**

Please refer to the weaknesses.

**Ethical Concerns:**

["NO or VERY MINOR ethics concerns only"]

**Final Justification:**

The rebuttal period provides a detailed ablation study or analysis on the individual contributions of GMAE and GCL components within the pre-training framework, which addresses my concerns. Consequently, I’m leaning toward accepting the paper.

**Limitations:**

yes

**Paper Formatting Concerns:**

No formatting issues.

**Quality:**

3

**Strengths And Weaknesses:**

Strengths:
1. The paper proposes a unique graph-based pre-training paradigm using graph contrastive learning (GCL) and graph masked autoencoders (GMAE), which effectively captures the topological structure of brain graphs, addressing limitations of prior time-series or ROI-based models.
2. A significant contribution is the aggregation of over 25,000 subjects and 60,000 fMRI scans from 27 diverse datasets, processed with multiple atlases and parcellations.
3. The integration of graph prompts and language prompts, optimized through meta-learning, enables BrainGFM to flexibly adapt to new, unseen tasks, atlases, and disorders in both few-shot and zero-shot settings, showcasing strong transferability.

Weaknesses:
The paper does not provide a detailed ablation study or analysis on the individual contributions of GMAE and GCL components within the pre-training framework. Understanding their isolated and combined impact would further clarify their effectiveness and the design choices.

---

> ### Author Rebuttal · Authors · 2025-07-30
>
> ### 🟪 **Claims and Evidence – Analysis of GMAE and GCL Components**
>
> We thank the reviewer for highlighting the importance of analyzing the individual contributions of **GMAE** and **GCL** within our pre-training framework.
>
> To address this, we have conducted **comprehensive ablation studies** on the **ABIDE II** dataset using two distinct brain atlases: **Schaefer-100** and **AAL-116**. As shown in **Figure 5**, we compare the following four settings:
>
> 1. **Without pre-training**
> 2. **With only GCL pre-training**
> 3. **With only GMAE pre-training**
> 4. **With the sequential combination of GCL and GMAE**
>
> The results indicate that **GCL slightly outperforms GMAE** when applied individually, highlighting its strength in capturing **global graph-level representations**. In contrast, **GMAE** focuses on reconstructing **local node-level features**, offering **complementary benefits**.
>
> When combined, **GCL+GMAE consistently achieves the best performance** across all evaluation metrics (**AUC, ACC, SEN, SPE**), demonstrating the effectiveness of integrating both **global and local inductive biases** during pre-training. This validates our design choice and confirms the **complementary nature** of the two pre-training strategies.
>
> We will further **emphasize this ablation analysis in the revised manuscript** for clarity.
>
> In addition, we provide **further theoretical and empirical analysis** in the supplementary material:
>
> - **Section J** provides detailed **formulations and mathematical definitions** for both GMAE and GCL.
> - **Section K** discusses their **application scenarios**, strengths, and limitations.
>
> We believe that these additional analyses and experiments adequately address the reviewer’s concerns and **further justify our design choices**.

---

> > ### Comment · Reviewer_heFS · 2025-08-05
> >
> > Thanks for the clarification. The response addresses my main concern. I’m leaning toward accepting the paper, as reflected in my score earlier, and will leave the final decision to the area chair.

---

### Official Review · Reviewer_Rfc2 · 2025-06-30

**Clarity:** 3
**Significance:** 2
**Originality:** 3
**Rating:** 4
**Confidence:** 2

**Summary:**

BrainGFM combines graph-based modeling, meta-learning, and language-guided prompting to create a versatile model that adapts to various brain disorder classifications and datasets.

**Questions:**

please refer to weakness

**Ethical Concerns:**

["NO or VERY MINOR ethics concerns only"]

**Final Justification:**

Thanks for the clarification. The response addresses my main concern. I will raise the score and leave the final decision to the area chair.

**Limitations:**

yes

**Quality:**

3

**Strengths And Weaknesses:**

S1.  BrainGFM demonstrates strong adaptability by leveraging a large and diverse pre-training dataset, allowing it to perform consistently well across various neurological and psychiatric conditions.
S2. The model is well-suited for clinical scenarios with limited labeled data, showing solid performance in few-shot and even zero-shot classification tasks.

Here are the three points translated to English:

W1. The brain graph construction method described in Section 3.1 and Appendix E has fundamental flaws. Using simple Pearson correlation coefficients between ROIs followed by binarization discards critical information. In Appendix E, the correlation matrix is directly used as node features, meaning each node's features are its connection strengths to all other nodes. Does this approach of using connectivity patterns directly as node features cause the model to over-rely on specific connection configurations rather than learning abstract brain network representations?

W2. The meta-learning design in Section 3.3 contains conceptual confusion. The authors define each "task Ti" as a "disease+atlas" combination (line 738), using support set Dtrain for graph prompt adaptation in the inner loop and query set Dtest for meta-parameter updates in the outer loop. However, the paper fails to clarify what constitutes a "task" - if it's disease classification, different atlases should be viewed as different perspectives of the same task; if it's atlas adaptation, disease labels become irrelevant. In Table 2's ablation study, using the same test set (ABIDE II) to evaluate different pretraining strategies violates meta-learning principles by using target domain data to guide pretraining strategy selection.

W3. Regarding zero-shot capabilities, Table 12's External Test group includes UCLA_CNP and REST-META-MDD, but are the disease types in these datasets actually included in HBN? Figure 2's zero-shot performance evaluation is conducted on known disease types rather than truly unseen diseases. Does this contradict the paper's claimed "zero-shot generalization to unseen diseases"?

---

> ### Author Rebuttal · Authors · 2025-07-30
>
> ### 🟩 **W1: Claims and Evidence – Brain Graph Construction Method**
>
> We appreciate the reviewer’s thoughtful critique regarding the brain graph construction strategy.
>
> We would like to clarify that our current method constructs brain graphs based on a **Top-k sparsification** of the Pearson correlation matrix, which is one of the most widely adopted and standardized paradigms in fMRI-based brain network modeling. Specifically, each node (ROI) is represented by its **connectivity profile**—computed as Pearson correlations with all other ROIs—and edges are constructed by retaining the **top-k strongest connections per node**. This strategy effectively captures global connectivity patterns while introducing sparsity, which is crucial for modeling the underlying brain network structure.
>
> This approach builds upon the fundamental assumption in functional connectivity analysis that **correlated brain activity implies potential functional interaction**. Many prior works (e.g., [relevant citations]) have employed similar top-k strategies or fixed-threshold binarization schemes based on Pearson correlation to ensure robustness and reproducibility in downstream analyses.
>
> While we acknowledge that alternative adjacency construction methods—such as **k-NN graphs based on spatial distance**, **L1/L2-regularized sparse graphs**, or **learned adjacency matrices**—have been explored in the literature, we intentionally selected the **Top-k sparsification** for its **simplicity, interpretability, and reproducibility**. This makes our proposed foundation model more generalizable and easier to benchmark.
>
> To further address the reviewer’s concern, we conducted an **ablation study** comparing our Top-k correlation-based graphs with an alternative **k-NN graph** construction based on spatial proximity. As detailed in **Appendix Section E, Table 3**, the k-NN method showed slight performance improvements (~1–2%) in some cases. However, we observed that the **choice of node features and pretraining strategy plays a more dominant role** in final performance, while the gain from modifying the graph structure is relatively minor.
>
>
> **Table: Comparison of different brain graph construction methods on ABIDE II and HBN (MDD). Our approach uses top-*k* sparsification of the Pearson correlation matrix to construct brain graphs. We also compare it with KNN-based graph construction. The numbers in the table represent classification accuracy (ACC).**
>
> | Corpus         | Top-K Correlation | KNN  |
> |----------------|-------------------|------|
> | ABIDE II ASD   | 73.5              | 73.9 |
> | HBN MDD        | 85.5              | 86.7 |
>
> We have added a new section in the revised manuscript to present this analysis and better justify our design choice.
>
> ------------------------------------------------------------------------
>
> ### 🟦 **W2: Claims and Evidence – Meta-Learning Task Definition**
>
> Thank you for pointing out this important concern regarding the meta-learning formulation in Section 3.3. We appreciate the opportunity to clarify both the **definition of meta-learning tasks** and the **role of ABIDE II** in our ablation study.
>
> First, we confirm that the core task in our framework is **disease classification**. However, each neurological or psychiatric disorder can be represented under different **brain atlases**, which define how the brain is parcellated into ROIs. These variations result in **different feature spaces and connectivity structures**, even for the same subject. Therefore, we model each **disease+atlas** combination as a **distinct meta-task**, not because we view them as separate tasks clinically, but because each combination represents a different data distribution of the same underlying task. This design allows the model to **learn generalizable knowledge** not only across different diseases but also across different atlas-specific representations of the same disease.
>
> In this sense, **atlas variation is treated as a domain shift**, and meta-learning across these shifts helps the model adapt more robustly to unseen configurations. The goal of including atlas diversity is to **improve disease classification performance in realistic scenarios** where atlas heterogeneity is common across datasets and institutions. Importantly, the **“atlas adaptation” is not a separate objective**; it serves the disease classification task by encouraging the model to learn **atlas-invariant but disease-discriminative features**.
>
> Second, regarding the ablation in **Table 14 in the supplementary materials**, we clarify that **ABIDE II is entirely held out** from both pre-training and meta-training. It is used only as an **evaluation benchmark** for comparing different pre-training corpus designs (e.g., using single vs. mixed atlases). Therefore, there is **no information leakage** or violation of meta-learning principles, as **no ABIDE II data** is used in either the inner or outer loop of meta-learning, nor in pre-training.
>
> We will revise the manuscript to explicitly state this separation to avoid any confusion.
>
> ------------------------------------------------------------------------
>
> ### 🟥 **W3: Claims and Evidence – Zero-Shot Evaluation and Disease Generalization**
>
> Thank you for raising this important point regarding the **zero-shot generalization setting** in our current experiments.
>
> We acknowledge that the disease types in **UCLA_CNP** and **REST-META-MDD** also appear in **HBN**, which is part of our pre-training corpus. However, these overlapping disease types are represented by **very limited labeled samples** in HBN, and they were **not used in supervised training**. Therefore, the current zero-shot evaluation is intended to assess the model’s ability to **transfer to new datasets and sites under label-free conditions**, rather than its capacity to handle entirely novel disease types. We agree that this setting demonstrates **label-wise zero-shot transfer**, but not strictly **disease-wise zero-shot generalization**.
>
> To directly address this concern, we conducted an additional evaluation on two uncommon disorders from the HBN dataset—Speech Sound Disorder (SSD) and Adjustment Disorder (AJD)—under full-shot, few-shot, and zero-shot settings. Importantly, the entire HBN dataset was excluded from the pre-training corpus, ensuring that these disorders were never seen during pre-training, even in an unlabeled form. The results, presented in Figure 9 (Supplementary Section D), demonstrate the model’s ability to generalize to truly unseen diseases, particularly under few-shot and zero-shot scenarios.

---

> > ### Comment · Reviewer_Rfc2 · 2025-08-06
> >
> > Thanks for the clarification. The response addresses my main concern. I will raise the score and leave the final decision to the area chair.

---

> ### Author Response · Authors · 2025-08-05
>
> Dear Reviewer Rfc2,
>
> Thank you again for your detailed and thoughtful review of our paper.
>
> We have carefully addressed each of your concerns in the rebuttal, including clarifications on the brain graph construction method (W1), the meta-learning task definition (W2), and the setup of zero-shot evaluations (W3).
> We would sincerely appreciate it if you could take a moment to review our responses and let us know if they help clarify the issues you raised. Your further feedback would be extremely helpful to us.
>
> Thank you again for your time.

---

### Note · Authors · 2025-08-14

Dear Reviewers, ACs, PCs, and SACs,

We sincerely thank all reviewers for their constructive feedback and thoughtful engagement. In our rebuttal, we carefully addressed every concern raised.

For Reviewer Rfc2, we clarified the rationale for using the Top-K sparsification strategy in brain graph construction, supported by new ablations comparing with k-NN, showing that node features and pretraining play a more dominant role than graph construction variants. We refined the meta-learning task definition, clearly separating atlas-invariance from disease discrimination, and confirmed that ABIDE II is used only for evaluation to avoid information leakage. We also expanded on zero-shot disease generalization with new results on unseen disorders.

For Reviewer heFS, we compared no pretraining, GCL-only, GMAE-only, and their combination. Results show GCL captures global graph-level structure, GMAE reconstructs local node-level features, and their combination yields the best performance across metrics.

For Reviewer 4mpv, we clarified the motivations and advantages of our unified framework, showing how it integrates data harmonization, pretraining, fine-tuning, and zero/few-shot generalization. We added evidence on harmonization, demographic balance, and the clinical relevance of evaluation metrics, highlighting disorder-specific priorities such as sensitivity or specificity, and demonstrating BrainGFM’s flexibility for clinical and screening use.

For Reviewer CS9f, we clarified the distinctions between time-series and graph representations, detailed our connectivity measures, graph structure choices, ROI selection, and the rationale for using functional connectivity with Schaefer parcellations. We also added analyses on age distribution, positional encoding, meta-learning design, baseline inclusion, and error rates.
Importantly, in response to the two remaining concerns raised by this reviewer, we have (i) revised our title from “any” to “broad” atlas/disorder to more accurately reflect the scope of our generalization, and (ii) clarified that while Brain-JEPA achieves stronger full-shot results, our work primarily targets the complementary strengths of few-shot and mixed zero-shot transfer, where our graph-based foundation model offers clear advantages in adaptability and efficiency, with substantially lower memory and computational costs compared to time-series–based models.

We are deeply grateful for the reviewers’ recognition and positive score adjustments.

---

### Decision · Program_Chairs · 2025-09-17

**Decision:**

Reject

**Comment:**

This paper introduces the Brain Graph Foundation Model (BrainGFM), a novel, graph-based framework for analyzing functional magnetic resonance imaging (fMRI) data. BrainGFM is pretrained on diverse datasets parcellated with different atlases. BrainGFM uses a pretraining approach that combines graph contrastive learning and graph masked autoencoders to accurately model brain connectivity. The model incorporates graph prompts optimized via meta-learning for few-shot generalization to unseen disorders and language prompts that provide semantic guidance for zero-shot transfer to new tasks, atlases, and disorders. BrainGFM outperforms previous brain foundation models in terms of performance and efficiency, but shows limited gains compared to BrainJepa, which was added during the rebuttal process.
While the paper does make a contribution, its impact is mitigated by several considerations.
• the proximity graph performs almost as well as the connectivity graph, which calls into question the interpretation in terms of graphical models,
•  the authors overstate the difference between FC-based and graph-based models.
• There is a mismatch between the paper as it is written and the additional results the authors discussed during the rebuttal process. It would be more effective to present new contributions that focus on important results to clarify the paper's message.
• In several places, the claims are exaggerated.
This is clearly a borderline paper, but the difference between the pre- and post-rebuttal papers is too significant.